# Cellular microRNA networks regulate host dependency of hepatitis C virus infection

Qisheng Li [1], Brianna Lowey[1], Catherine Sodroski[1], Siddharth Krishnamurthy[1], Hawwa Alao[1], Helen Cha[1], Stephan Chiu[1], Ramy El-Diwany[1], Marc G. Ghany[1] & T.Jake Liang[1]

Cellular microRNAs (miRNAs) have been shown to regulate hepatitis C virus (HCV) replication, yet a systematic interrogation of the repertoire of miRNAs impacting HCV life cycle is lacking. Here we apply integrative functional genomics strategies to elucidate global HCV–miRNA interactions. Through genome-wide miRNA mimic and hairpin inhibitor phenotypic screens, and miRNA–mRNA transcriptomics analyses, we identify three proviral and nine antiviral miRNAs that interact with HCV. These miRNAs are functionally linked to particular steps of HCV life cycle and related viral host dependencies. Further mechanistic studies demonstrate that miR-25, let-7, and miR-130 families repress essential HCV cofactors, thus restricting viral infection at multiple stages. HCV subverts the antiviral actions of these miRNAs by dampening their expression in cell culture models and HCV-infected human livers. This comprehensive HCV–miRNA interaction map provides fundamental insights into HCV-mediated pathogenesis and unveils molecular pathways linking RNA biology to viral infections.

[1] Liver Diseases Branch, National Institute of Diabetes and Digestive and Kidney Diseases, National Institutes of Health, Bethesda, MD 20892, USA. Correspondence and requests for materials should be addressed to Q.L. (email: liqisheng@niddk.nih.gov) or to T.J.L. (email: jakel@bdg10.niddk.nih.gov)

MicroRNAs (miRNAs) are endogenous, ~22-nt non-coding RNAs that exert important regulatory roles by pairing to the 3′-untranslated regions (UTRs) of target mRNAs leading to their repression[1]. Mature miRNAs are derived from primary Pol II-transcribed RNA precursors, through sequential cleavages by Drosha and Dicer. In the presence of the RNA-induced silencing complex (RISC), miRNAs regulate gene expression via both translational repression and mRNA degradation[2]; destabilization of mRNAs appears to be the predominant mechanism that mediates target inhibition[3]. miRNAs are projected to target the majority of mRNAs in the human genome[4], thus participating in a wide range of biological processes such as

development, immunity, cancer, and pathogen infections. miRNA themselves are also regulated in a highly tissue-specific manner, while altered miRNA expression patterns have been linked to various human diseases.

Host miRNAs may target viral genomes or cellular factors, positively or negatively regulating viral infection[5]. Viral infections on the other hand can impact cellular miRNA expression levels or co-opt miRNA-mediated pathways, to alter cellular functions and create a favorable environment for their survival and pathogenic effects[5, 6]. Hepatitis C virus (HCV), a hepatotropic RNA virus of the *Flaviviridae* family, infects ~170 million people worldwide. Chronic hepatitis C often progresses to end-stage liver diseases, including cirrhosis and hepatocellular carcinoma[7]. Recent studies have revealed that HCV relies heavily on cellular factors throughout its life cycle[8–13]. Cellular miRNAs, such as miR-122 and miR-196a represent important HCV host dependencies through their actions either directly on the viral genome or indirectly on virus-associated host factors to modulate viral infection[14]. Antagonism of miR-122 with an antisense oligonucleotide results in long-lasting suppression of HCV viremia in chimpanzees and humans, conferring a therapeutic strategy against chronic HCV infection[15, 16].

In this study, we performed high-throughput screens employing the whole-genome miRNA mimics and hairpin inhibitors and investigated their functions. We uncovered multiple miRNAs as regulators of HCV infection, including the miR-25, let-7, and miR-130 families. Next, through transcriptome-wide miRNA target prediction, identification, and validation, we elucidated an exhaustive functional map that illustrates interactions among cellular miRNAs, target mRNAs, and HCV throughout the viral life cycle. miRNA profiling analyses revealed that HCV modulates the expression of various cellular miRNAs to overcome host antiviral restrictions and thus promote viral propagation and persistence in the liver.

## Results

**Genome-wide miRNA functional screen.** To analyze global HCV–miRNA interactions, we performed a genome-wide miRNA functional screen using libraries of synthetic, chemically modified mimics and hairpin inhibitors. The combined miRNA agonist and antagonist screen assessed the impacts of all mature miRNAs on both the early (part-one) and late (part-two) stages of HCV infection. The schematic of the screen is illustrated in Fig. 1a and has been successfully applied to our earlier genome-wide small interfering RNA (siRNA) screen for HCV host dependencies[8]. Detailed screening protocols are described in Methods, and the primary mimic and hairpin inhibitor screen results are summarized in Supplementary Data 1 and 2, respectively. Seventy-two miRNAs in the mimic library were excluded from hit selection and further analysis due to cytotoxicity (Supplementary Data 1, and explained in Methods).

**miRNA phenotypic and bioinformatics analysis.** All miRNAs studied in the primary screen are shown in volcano plots to display their relevance to HCV infection. Hits were defined as those with an average Z score of either greater than 1 (proviral) or

less than −1 (antiviral), and a P value <0.05 (Student's t-test) (Fig. 1b–e; Supplementary Fig. 1; Supplementary Data 1 and 2). With these criteria, we identified 276 miRNA mimics and 153 hairpin inhibitors as primary hits (Supplementary Data 1 and 2). These hits were cross-referenced resulting in a list of 31 miRNAs that demonstrated opposite phenotypes in the gain-of-function and loss-of-function assays (e.g., proviral effect by mimic and antiviral effect by inhibitor or vice versa) (Fig. 1f). This approach identified 10 miRNAs that enhance HCV infection and 21 with antiviral effects, and includes previously known HCV-associated miRNAs such as miR-122 and miR-196a (Fig. 1f). The strict selection criterion of opposite phenotype in both the mimic and inhibitor screens may exclude hits that could still influence the HCV life cycle (see Discussion).

Notably, there was a stark difference in hits distribution and magnitude of activities between the mimics and hairpin inhibitors. The inhibitor screen yielded fewer hits compared to the mimics under the same selection criteria (Fig. 1b–e, Supplementary Fig. 1 and Supplementary Data 1 and 2). This difference in general phenotypes between inhibitors and mimics was also observed in other miRNA screens[17], and can be partially explained by the fact that hairpin inhibitors only mask the functions of expressed endogenous miRNAs, while not all miRNAs are abundantly expressed in any given cell type. miRNA inhibition may also show a weaker effect due to the compensatory effects of other miRNAs in the same family—miRNAs that belong to the same family execute similar physiological functions, therefore targets of one miRNA are subject to regulation by all other family members[1].

**miRNA transcriptomics analysis.** To define whether the miRNAs identified from the screen are physiologically relevant in hepatocytes, we analyzed miRNA transcriptomes in both Huh7.5.1 cells and primary human hepatocytes (PHHs) by performing nCounter miRNA expression assays using NanoString technology (Supplementary Fig. 2a, b; Supplementary Data 3 and 4). The NanoString nCounter platform enables highly multiplexed, single-molecule digital counting of miRNA transcripts with high precision and sensitivity[18]. The abundance of each mature miRNA was normalized as miRNA reads (counts per cell) (Supplementary Data 3 and 4). In Huh7.5.1 cells, 82.25% miRNAs are barely expressed (<50 counts, 658 miRNAs), 12.38% are modestly expressed (50–300 counts, 99 miRNAs), and only 5.38% are highly expressed (>300 counts, 43 miRNAs) (Supplementary Fig. 2a, c). These highly expressed miRNAs accounted for 76.64% of total reads (Supplementary Fig. 2a, c). In PHHs, the miRNA expression landscape is highly similar to that in Huh7.5.1 cells (Fig. 2a; Supplementary Fig. 2b, d). As expected, miR-122 is the most abundantly expressed miRNA in the liver miRNome (Fig. 2a).

To further probe global miRNA expression in Huh7.5.1 cells, we conducted microarray analysis. The microarray-based transcriptomic profiling of cellular miRNAs, while less sensitive and quantitative as the NanoString technology, generated consistent data with those identified by the nCounter platform (Supplementary Fig. 3; Supplementary Data 5). In addition, the general

**Fig. 1** Integrative functional screens identify cellular miRNAs modulating HCV infection. **a** Schematic of the primary screen. **b–e** Volcano plots showing Z scores and P values ($-\log_2$ scale, Student's t test) from miRNA mimics in part one (**b**) and part two (**c**), and inhibitors in part one (**d**) and part two (**e**) of the HCVcc screen. Representative images and quantitative analyses of HCV core staining in cells transfected with select miRNA mimics or inhibitors, as indicated, are shown beside the volcano plots. Green, HCV core; blue, cell nuclei. Scale bars, 100 μm. Numbers represent the percentages of core staining-positive cells (the mean ± SD, n = 3). Ctrl, miRNA mimic or inhibitor negative control. **f** Heatmap eliciting Z scores for primary screen hits with opposite mimic and inhibitor phenotypes. Z scores in the heatmap are depicted in a continuum from blue (reduced infection—antiviral) to red (enhanced infection—proviral)

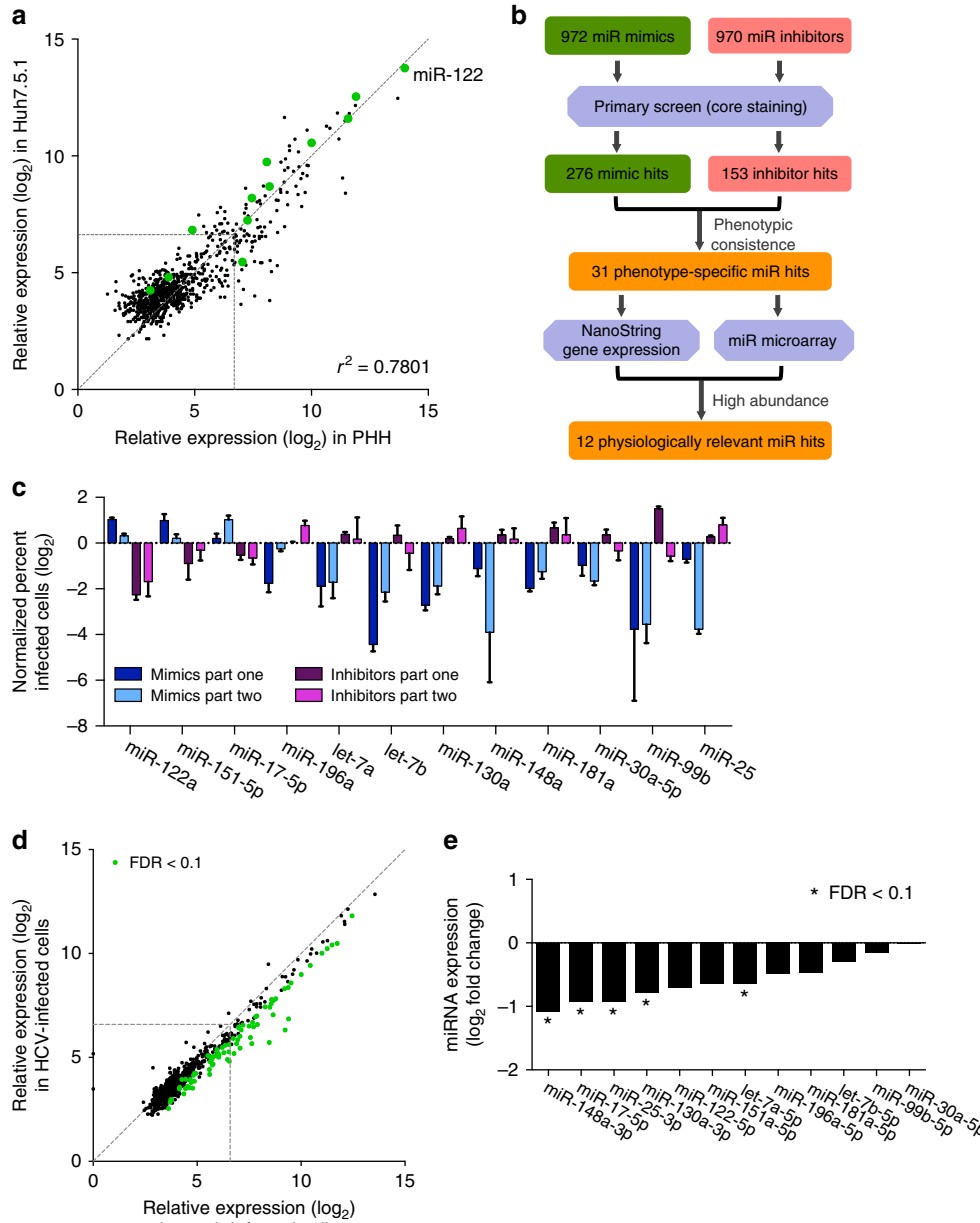

**Fig. 2** HCV infection downregulates miRNA expression landscape. **a** miRNA transcriptome analysis using NanoString technology. Graph illustrates normalized transcript reads in Huh7.5.1 cells and primary human hepatocytes (PHH) ($\log_2$ scale). Validated miRNA screen hits are depicted in green. $X$–$Y$ dashed line represents 0.1% abundance in the nCounter assays. **b** Schematic diagram outlining workflow for identification of physiologically relevant miRNA screen hits. **c** Phenotypes of 12 validated miRNA hits in impacting productive HCV infection. Normalized percent-infected cells for miRNA mimics or inhibitors comparing to negative controls are shown ($\log_2$ scale). Error bars represent SD of the mean, $n = 3$. **d** HCV infection generally reduces miRNA expression in Huh7.5.1 cells. Relative expression level of each miRNA represents average of 24, 48, and 72 h of HCV infection or mock treatment. miRNAs significantly downregulated by HCV infection (Student's $t$-test, FDR < 0.1) are shown in green. **e** HCV infection downregulates the expression of biologically relevant cellular miRNAs. Graph depicts fold change comparing infected (average of 24, 48, and 72 h) and uninfected cells ($\log_2$ scale). *FDR < 0.1

miRNA expression profiles, particularly distribution of abundant miRNAs described here, are similar to other hepatic miRNA transcriptome data sets[19, 20].

**Abundantly expressed miRNA hits are dysregulated by HCV.** We next evaluated whether the 31 phenotype-specific miRNA hits (Fig. 1f) are physiologically relevant to HCV infection by probing their hepatocellular expression across NanoString and microarray data sets (Fig. 2b). Previous studies have suggested a

cellular threshold of miRNA level >100 reads per million (>0.1% abundance) as necessary to exert a regulatory effect on its targets[21]. Twelve miRNA hits, each accounting for more than 0.1% in total reads (around 100,000 in both Huh7.5.1 cells and PHHs) when quantified by either nCounter or microarray approach, meet this threshold (Supplementary Fig. 4). We reason that these hits are physiologically relevant miRNAs interacting with HCV and fine-tune its infection. Among them, three are proviral miRNAs (miR-122, miR-151-5p, and miR-17-5p), and nine others, including let-7a, let-7b, miR-130a, miR-148a, miR-181a,

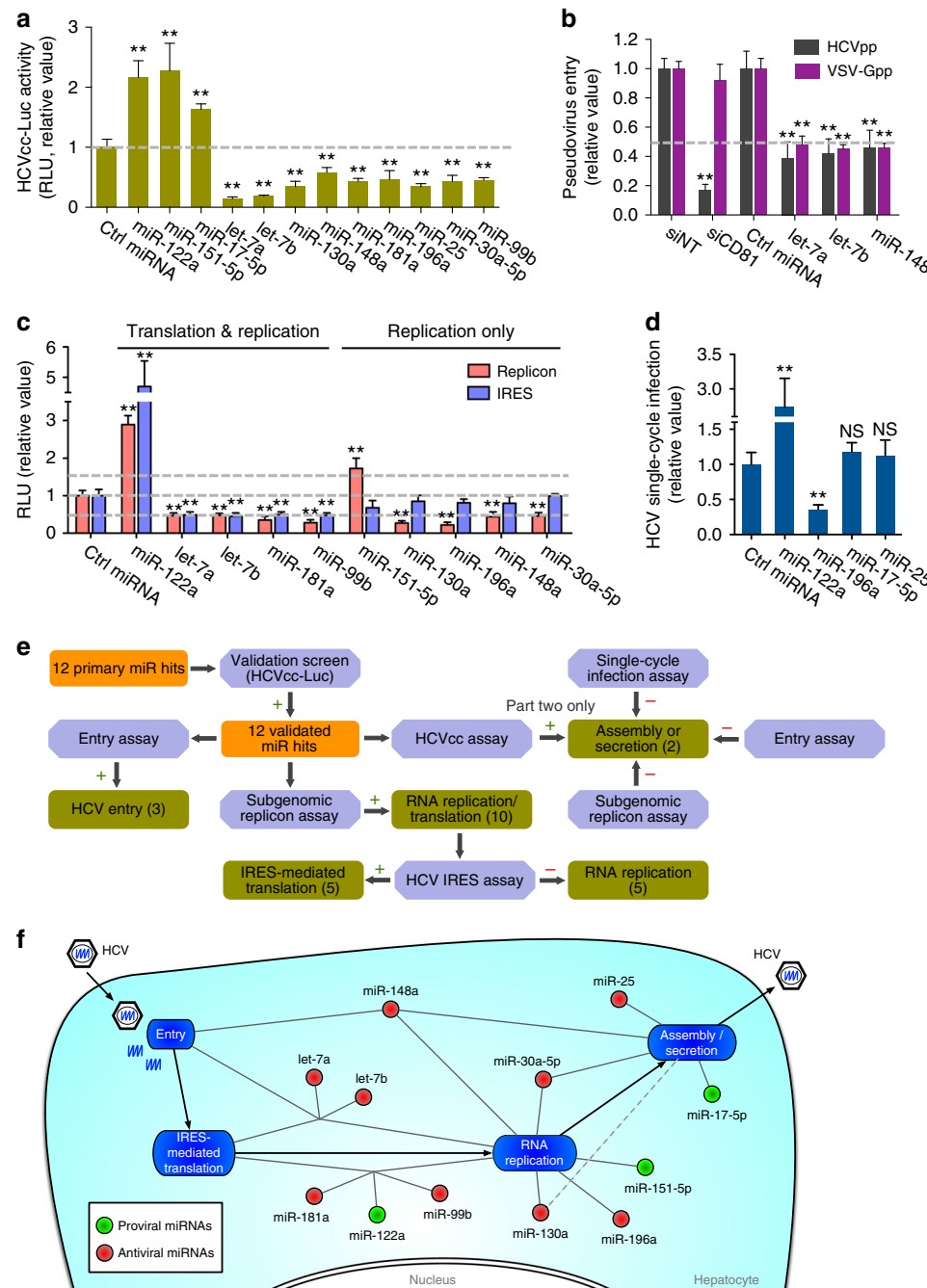

**Fig. 3** Impacts of cellular miRNAs on HCV life cycle. **a** Secondary screen using HCVcc-Luc confirmed the phenotypes of primary screen hits in HCV propagation. Relative luciferase activities upon treatment of various miRNA mimics are shown. **b** Identification of host miRNAs modulating HCV entry. Graphs show the effects of indicated miRNA mimics on entry of luciferase-encoding pseudotyped viruses bearing HCV (HCVpp) or VSV (VSV-Gpp, serving as a control) envelope proteins. **c** Identification of miRNAs regulating HCV IRES-mediated translation or RNA replication. HCV subgenomic replicon and IRES assays were conducted, and miRNA hits exerting significant effects on HCV RNA replication and/or translation are shown. **d** HCV single-cycle infection assay (HCVsc) recapitulates viral entry, translation, and replication, but not assembly or secretion of virions. **a**–**d** Values are normalized relative to the mimic control (set as 1), and error bars represent SD of the mean, $n = 5$. **$P < 0.01$ determined by Student's $t$-test. **e** Schematic flow diagram of identifying cellular miRNAs associated with various HCV life cycle stages. **f** Integrated map depicting interactions of host miRNAs with the entire HCV life cycle. The framework was constructed based on the sequence of HCV infection steps from entry to assembly/secretion. Proviral miRNAs are shown as green circles, and antiviral miRNAs are shown as red circles. The dashed gray line indicates a link between miR-130a and the viral assembly/secretion step identified in later mechanistic studies (Fig. 6)

miR-196a, miR-30a-5p, miR-99b, and miR-25, are antiviral factors (Fig. 2c).

Multiple host miRNAs that modulate HCV infection have been recently uncovered[14]. We asked whether HCV infection impacts the miRNA landscape as a strategy to subvert host stress responses. We infected Huh7.5.1 cells with HCV for 24, 48, and 72 h and examined global miRNA expression through NanoString nCounter analysis. HCV infection exerts a broad repression of cellular miRNAs at all time points (Fig. 2d; Supplementary Fig. 5a). Particularly, 78 out of the 142 highly

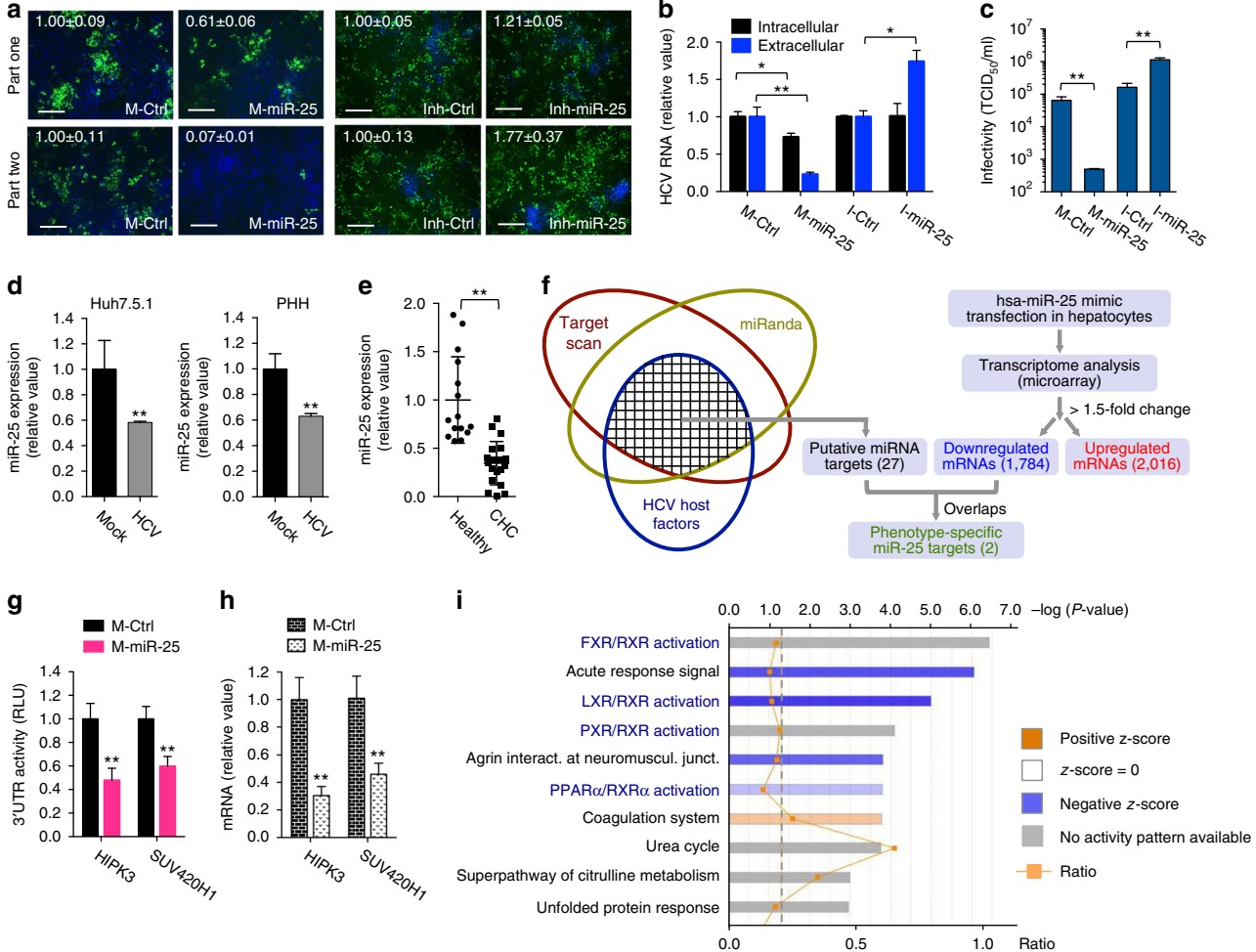

**Fig. 4** miR-25 restricts the late stages of HCV infection. **a–c** Effects of miR-25 mimic or hairpin inhibitor on production of HCV core protein (**a**), viral RNA (**b**), and infectious HCV (**c**). The mimic and inhibitor exerted opposite phenotypes in Huh7.5.1 cells. Compared with part-one core staining (**a**) and intracellular HCV RNA levels (**b**), more profound inhibitory effects were seen in part-two core staining (**a**) and in the level of secreted HCV RNA (**b**). (**a**) Green, HCV core; blue, nuclei. Scale bars, 100 μm. (**d**) HCV infection downregulates miR-25 expression in Huh7.5.1 cells and PHH, examined by qPCR gene expression assays. **e** Hepatic abundance of miR-25 is significantly lower in the livers of CHC patients compared to those of healthy controls, measured by qPCR. Each dot represents one individual's liver tissue. **f** Schematics for systematic identification of potential miR-25 targets. Bioinformatics-based target prediction, in line with assessing phenotypic effects on HCV infection and global transcriptome analysis, derived two phenotypic-specific miR-25 target candidates. **g, h** miR-25 mimic transfection abates 3′-UTR activities (**g**) and mRNA levels (**h**) of HIPK3 and SUV420H1. **i** Bioinformatics analysis of miR-25 targeted molecular pathways, derived from microarray-based transcriptome dataset comparing miR-25 mimic-transfected cells with mimic control-treated cells (Supplementary Data 7). Ingenuity Pathway Analysis (IPA, QIAGEN) was implemented to analyze all genes significantly downregulated by miR-25 mimic treatment, identifying various enriched pathways, as shown, that significantly correlate with miR-25 overexpression. All values are normalized relative to Ctrl, and error bars represent SD of the mean, $n = 3$ (**a–d**, **h**) or 5 (**g**). *$P < 0.05$, **$P < 0.01$ determined by Student's $t$-test

(>300 counts) or moderately expressed miRNAs (50–300 counts) were downregulated by more than 2-fold upon HCV infection. miR-1246 and six others were upregulated by more than 1.5-fold at any time point (Supplementary Fig. 5b). PHHs exhibited similar, though to a lesser extent, dysregulation of miRNA expression in HCV-infected cells (Supplementary Data 4); this difference is likely due to insufficient viral propagation in primary cells. Intriguingly, out of the 12 validated miRNA hits, five (miR-148a, miR-17-5p, miR-25, miR-130a, and let-7a) were significantly downregulated by HCV, while others were also suppressed, but to a lesser extent (Fig. 2e). These results indicate cellular miRNAs both regulate and are regulated by HCV infection.

**Functional deconvolution of miRNAs in the HCV life cycle**. To validate the impacts of the 12 primary hits that are abundantly

expressed in hepatocytes, we performed a secondary screen using an HCVcc model that encodes a luciferase reporter (HCVcc-Luc). A validated hit was defined as one that exhibited consistent proviral or antiviral phenotype to that demonstrated in the primary screen, and exerted at least a 1.5-fold change in HCVcc-Luc infection compared with the negative control, with a $P$ value <0.01 (Student's $t$-test). All three proviral and nine antiviral miRNAs were confirmed in modulating HCV infection (Fig. 3a).

To dissect the exact functions of these 12 HCV-associated miRNAs and evaluate their relevance to different stages of HCV life cycle, we interrogated each miRNA using various HCV in vitro models[9] (Fig. 3b–d). The flowchart of the life cycle assays is illustrated in Fig. 3e.

First, we performed viral entry assays to examine the effects of the 12 miRNAs on HCV pseudoparticle (HCVpp) or vesicular stomatitis virus G protein pseudoparticle (VSV-Gpp) entry. Overexpression of let-7a, let-7b, or miR-148a significantly

blocked both HCVpp and VSV-Gpp infections (Fig. 3b). Thus, these miRNAs not only impede HCV entry, but potentially influence viral entry more broadly.

By performing HCV subgenomic replicon assays that assess viral RNA replication and protein translation, we identified let-7a, let-7b, miR-181a, and miR-99b as restriction factors impacting either HCV IRES-mediated translation specifically or both translation and replication (Fig. 3c). Five miRNAs targeted the replication stage only, as their overexpression significantly enhanced (miR-151-5p) or diminished (miR-130a, miR-196a, miR-148a, and miR-30a-5p) HCV RNA replication but not IRES-mediated translation in the replicon assays (Fig. 3c). miR-122 enhanced these steps as expected.

Furthermore, through the two-part HCVcc infection assays, we uncovered miR-17-5p (proviral) and miR-25 (antiviral) as host dependencies that preferentially regulate the late stage of HCV infection—assembly or secretion. Transfection of both miRNAs in hepatocytes exerted greater effects on part-two core than part-one core staining, indicating interference in the later steps of HCV life cycle[8]. To confirm these findings, we conducted single-cycle infection assays (HCVsc) using a core-defective, HCV trans-packing system that only generates single-round infectious virus, thus distinguishing early stages of viral life cycle from late stages[9]. We showed that unlike miR-122 and miR-196a, which predominantly act on the early stages of HCV infection, miR-17-5p or miR-25 had no effect on HCVsc infection (Fig. 3d), validating their roles in the late stages only.

Collectively, we generated an integrated HCV–miRNA interaction map that displays 12 HCV-associated cellular miRNAs and their proviral or antiviral roles in particular steps of the viral life cycle (Fig. 3f).

**miR-25 impedes HCV assembly or secretion.** Next we dissected the functions and mechanisms of three miRNAs: miR-25, let-7a, and miR-130a in modulating HCV infection. miR-25 targets the late steps of HCV infection as overexpression of miR-25 in Huh7.5.1 cells drastically diminished HCV core protein production in part two (Fig. 4a). miR-25 mimic transfection also led to a more profound inhibitory effect on the secreted HCV RNA level than that of intracellular viral RNA (~4-fold vs. ~1.5-fold in reduction) (Fig. 4b). Moreover, production of infectious HCV in the supernatant was inhibited by overexpressing miR-25 (Fig. 4c), the level of which was markedly increased by transfection of its mimic (Supplementary Fig. 6a). Infection of Huh7.5.1 cells with shMIMIC lentiviral miR-25 inhibited HCV infection in a lentiviral MOI-dependent manner (Supplementary Fig. 6b, c), confirming miR-25's gain-of-function effect. On the contrary, transfection of miR-25 hairpin inhibitor significantly elevated part-two core protein expression, level of extracellular viral RNA, and infectious HCV production (Fig. 4a–c). These data support miR-25's role in the late stages of HCV life cycle (assembly or secretion).

HCV infection markedly decreased miR-25 expression in hepatocytes, as shown in Fig. 2e and Supplementary Fig. 5. We conducted qPCR assays and confirmed HCV-mediated down-regulation of miR-25 in both Huh7.5.1 cells and PHHs (Fig. 4d). In the liver tissues of chronic hepatitis C (CHC) patients, miR-25 expression level was significantly lower than those of normal livers (Fig. 4e). miR-25 expression seems to be unrelated to the extent of liver fibrosis—assessed by Ishak score (Supplementary Fig. 6d). These results suggest a previously unidentified viral escape mechanism by disrupting the antiviral effects of cellular miRNAs (a legitimate host restricting strategy) to achieve effective and persistent infection in human liver.

To gain further insight into the biological activities of miR-25, we surveyed cellular targets through which miR-25 may deploy its antiviral functions. First, we performed bioinformatics analyses to identify putative target mRNAs using two miRNA target prediction tools, TargetScan[22] (www.targetscan.org) and miRanda-mirSVR[23] (www.microrna.org). The two bioinformatics tools are complementary rather than redundant; thus, by overlapping the predicted target genes generated from both algorithms, we could predict miRNA targets with greater confidence. 27 of the predicted miR-25 targets were previously linked to HCV infection[9], and thus were selected for validation (Fig. 4f). For all 12 HCV-associated miRNAs, the lists of computationally predicated cellular targets which are known HCV host factors are shown in Supplementary Data 6.

Since miRNAs regulate gene expression predominantly through mRNA degradation[3], we performed a microarray analysis and assessed global transcriptome changes in Huh7.5.1 cells transfected with miR-25 mimic. The microarray identified 1784 downregulated and 2016 upregulated mRNAs, all exceeding a 1.5-fold change (Fig. 4f; Supplementary Data 7). Cross-referencing these hits with the 27 predicted targets and HCV host factors revealed two putative phenotype-specific targets: homeodomain interacting protein kinase 3 (HIPK3) and suppressor of variegation 4–20 homolog 1 (SUV420H1) (Fig. 4f; Supplementary Fig. 7a). HIPK3 is a protein kinase that regulates mRNA transcription and Fas-mediated apoptosis, and SUV420H1 is a relatively uncharacterized gene that may play a role in histone methylation. Bioinformatics analyses demonstrated that miR-25 possesses a seed sequence match site at the 3′ UTR of each gene (Supplementary Fig. 7b). To evaluate whether miR-25 targets these host factors, we cloned the entire 3′ UTR of each gene downstream of a luciferase reporter and performed 3′ UTR assays (Fig. 4g; Supplementary Fig. 7b). We showed that HIPK3 and SUV420H1 3′-UTR activities were significantly inhibited by miR-25 mimic transfection, indicating that these 3′-UTRs are subject to miR-25 regulation (Fig. 4g). We further confirmed by qPCR that miR-25 overexpression reduced their mRNA levels in Huh7.5.1 cells (Fig. 4h), consistent with the microarray results (Supplementary Data 7). However, we cannot completely rule out that miR-25 may regulate HIPK3 and SUV420H1 in an indirect manner.

Both HIPK3 and SUV420H1 are involved in the late stage of HCV infection[9]. Depletion of either gene by siRNA drastically decreased the extracellular level of HCV RNA, but had no or little effect on the intracellular level, nor on the replicon or HCVsc activity (Supplementary Fig. 7c–f). Thus, miR-25 selectively inhibits the expression of two essential HCV host factors, HIPK3 and SUV420H1, to block HCV assembly and production (Supplementary Fig. 7g). Interestingly, microarray-based transcriptome analysis of miR-25 mimic-treated cells also demonstrated that there is a significant enrichment of nuclear receptor activation pathways, including FXR/RXR, LXR/RXR, PXR/RXR, and PPARA/RXR pathways (Fig. 4i). These nuclear receptors may regulate various proviral or antiviral signaling (such as metabolism) to affect HCV infection in hepatocytes[24].

**let-7a inhibits HCV infection at multiple steps.** Two members of the let-7 family, let-7a, and let-7b, restrict multiple steps of the HCV life cycle—entry, translation, and RNA replication (Fig. 3f). Among all let-7 family members, let-7a is expressed most abundantly in hepatocytes (Supplementary Fig. 8a, b). let-7a overexpression by transfecting its mimic or infection of Huh7.5.1 cells with shMIMIC lentiviral miRNA significantly reduced core production expression in both part-one and part-two HCVcc assays (Fig. 5a; Supplementary Fig. 8c, d), whereas overexpression of let-7a hairpin inhibitor led to an opposite effect (Fig. 5a). let-7a mimic transfection also reduced HCV RNA production and viral infectivity (Fig. 5b–d and Supplementary Fig. 8e), conforming an antiviral role of let-7a in hepatocytes. Meanwhile, HCV infection

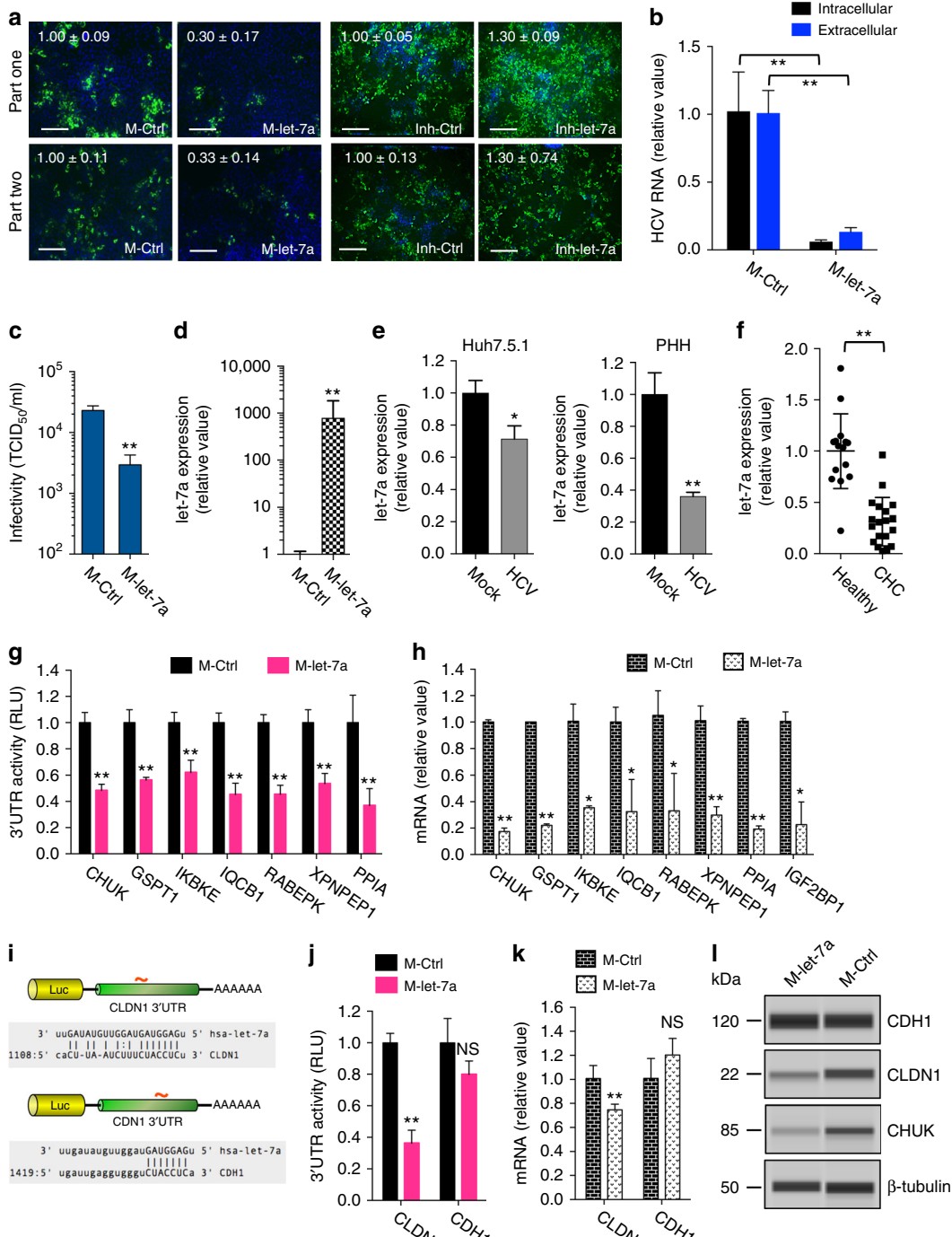

**Fig. 5** let-7a targets various HCV host dependencies to inhibit multiple steps of HCV infection. **a–c** Effects of let-7a mimic transfection on productive HCV infection. **a** Representative images and quantitative analyses of HCV core staining in Huh7.5.1 cells transfected with mimic control (M-Ctrl) or let-7a, for part one and part two of the screen. Green, HCV core; blue, nuclei. Scale bars, 100 μm. **b** Quantification of intracellular and extracellular HCV RNA levels after let-7a mimic treatment. **c** let-7a mimic transfection suppresses HCV infectivity, demonstrated by TCID$_{50}$ assay. **d** Transfection efficiency of let-7a mimic, determined by Q-PCR. **e** HCV infection decreases let-7a expression in Huh7.5.1 cells and PHH. **f** Hepatic expression levels of let-7a are significantly lower among CHC patients compared to healthy controls, measured by Q-PCR. Dots represent assessed individuals. **g, h** Validation of let-7a cellular targets. 3′-UTR activities (**g**) and mRNA levels (**h**) of various indicated targets were shown downregulated by let-7a mimic transfection. **i–l** let-7a specifically represses CLDN1 expression to inhibit HCV entry. **i** CLDN1 and CDH1 each contain one predicted let-7a binding site in the 3′-UTR. **j** Effect of let-7a transfection on CLDN1 and CDH1 3′UTR activities. **k** Effect of let-7a transfection on CLDN1 and CDH1 mRNA levels. **l** Effects of let-7a transfection on protein levels of various indicated targets, determined by western blot. β-tubulin was used as a loading control. kDa kilo Daltons. Values are normalized relative to M-Ctrl, and error bars represent SD of the mean, $n = 3$ (**a–e**, **h**, **k**) or 5 (**g**, **j**). *$P < 0.05$, **$P < 0.01$ determined by Student's $t$-test. NS not significant

significantly suppressed let-7a expression in cultured hepatocytes including Huh7.5.1 cells and PHHs, as well as in liver biopsy tissues from CHC patients (Fig. 5e, f). The extent of liver fibrosis, assessed by Ishak score, did not affect let-7a expression levels (Supplementary Fig. 8f).

To identify the cellular targets of let-7a that may mediate its antiviral effects, we applied the same algorithm employed for defining miR-25 targets (Supplementary Fig. 9a). We uncovered eight HCV host factors as putative let-7a targets, including CHUK or IKK-α, a major IκB kinase in the activation of the non-canonical NF-κB pathway; another non-canonical IκB kinase IKBKE; cyclophilin A (PPIA); the calmodulin (CaM) binding protein IQCB1; the insulin-like growth factor 2 mRNA-binding protein (IGF2BP1); RABEPK, a Rab9 effector protein implicated in vesicle docking and the trans-Golgi network; the aminopeptidase XPNPEP1; and the GTPase and GTP-binding protein GSPT1 (Supplementary Fig. 9b). Each of these genes has one or more let-7a seed sequence sites in its 3′-UTR (Supplementary Fig. 9c). We performed 3′-UTR assays to assess whether let-7a targets these predicted binding sites. Transfection of let-7a mimic significantly inhibited the luciferase activity of each 3′-UTR construct (Fig. 5g), suggesting let-7a may act on these 3′-UTRs. Furthermore, let-7a overexpression in Huh7.5.1 cells markedly reduced the mRNA levels of all eight genes (Fig. 5h).

Previous studies suggest these let-7a targets might affect different stages of HCV life cycle[9]. We used siRNAs to knock down these genes in hepatocytes and performed HCVcc and replicon assays to confirm their phenotypes (Supplementary Fig. 9d, e). IGF2BP1, IQCB1, and RABEPK were shown to affect HCV IRES-mediated translation, while GSPT1 and PPIA modulate HCV RNA replication (Supplementary Fig. 9e). Interestingly, three other validated let-7a targets, CHUK, IKBKE, and XPNPEP1, preferentially act on HCV assembly or secretion[9] (Supplementary Fig. 9d); implying that let-7a also acts on the late stage of HCV life cycle through targeting CHUK, IKBKE, and XPNPEP1 (Supplementary Fig. 9f). The initial assays (Fig. 3) did not designate let-7a to the assembly/secretion steps, because the algorithm for defining late-stage factors depends largely on the predominant HCVcc part-two phenotype, thereby missing the late step-associated miRNAs that also demonstrate a strong phenotype in the early steps, such as let-7a (Fig. 5a, b).

Surprisingly, none of the above-identified let-7a targets encode a viral entry factor; thus a mechanistic link between let-7a and the observed inhibitory effect on HCV entry is still missing. We scanned the mRNA sequences of all known HCV entry factors and identified a let-7a seed matching site in the 3′UTR of claudin-1 (CLDN1)[25] and E-cadherin (CDH1)[26] (Fig. 5i). These entry factors were excluded from the initial analysis because the microarray-based mRNA quantification did not show a decrease in their mRNA levels upon let-7 mimic transfection. Given the imperfect nature of high-throughput microarray analysis and the possibility of miRNA regulation of only protein translation, we further evaluated CLDN1 and CDH1. let-7a overexpression in hepatocytes significantly suppressed the 3′-UTR activity of CLDN1 but not that of CDH1 (Fig. 5j). Gene expression assays showed that let-7a mimic only slightly decreased CLDN1 mRNA level and had no effect on CDH1 mRNA level (Fig. 5k). CLDN1 protein level, nevertheless, was drastically decreased upon let-7a overexpression, comparable to that of another confirmed let-7a target, CHUK (Fig. 5l). Therefore, we demonstrated that let-7a targets CLDN1 and represses its translation to block HCV entry (Supplementary Fig. 9f).

**let-7 family of miRNAs collectively restrict HCV propagation.** In our screens, miRNAs belonging to the same family manifested

a comparable HCV-regulating phenotype. Eight of the nine let-7 family members—let-7a, let-7b, let-7c, let-7e, let-7f, let-7g, let-7i and miR-98—were identified as antiviral hits from the primary mimic screen (Supplementary Data 1). Of note, let-7d was excluded from initial hit selection due to apparent cytotoxicity induced by its mimic, although a strong anti-HCV activity was observed (Supplementary Fig. 10a, Supplementary Data 1). Transfection of each let-7 mimic significantly inhibited HCV core and RNA production in hepatocytes (Supplementary Fig. 10a, b). The addition of individual let-7 hairpin inhibitors had only a small effect on HCV infection in the primary screen (Supplementary Data 2), likely due to the other let-7 family members still present within the cells, which may compensate for the inhibition of only one miRNA. Addition of all let-7 family hairpin inhibitors in combination, however, considerably increased HCV RNA production (Supplementary Fig. 10c). The nine let-7 miRNAs share an identical seed sequence (Supplementary Fig. 10d, seed regions are shown in blue), suggesting they target the same set of cellular factors. We examined two validated let-7a targets, CHUK and PPIA, and showed that their 3′-UTR activities were inhibited by each of other let-7 miRNAs (Supplementary Fig. 10e, f). Similarly, the protein levels of CHUK and CLDN1 were repressed by these let-7 family members (Supplementary Fig. 10g).

**miR-130 miRNAs suppress HCV replication and assembly.** In our primary screen, two major miR-130 miRNAs, miR-130a-3p and miR-130b-3p (denominated as miR-130a/b hereafter), and another family member, miR-301a, were shown to effectively inhibit HCV infection (Supplementary Fig. 11a). Viral life cycle assays indicate that miR-130a, the family member with the highest expression level in hepatocytes (Supplementary Fig. 11b), preferentially interferes with HCV RNA replication (Fig. 3f). Transfection of miR-130a synthetic mimic in Huh7.5.1 cells significantly decreased core protein and HCV RNA expression (Fig. 6a–c; Supplementary Fig. 11c). Similarly, the lentiviral vector-based miR-130a overexpression inhibited HCV infection in both part-one and part-two assays (Supplementary Fig. 11d, e). In contrast, blocking miR-130a functions with its hairpin inhibitor markedly enhanced HCV production (Fig. 6a, b). These results confirmed a crucial role of miR-130a in restricting HCV infection. Interestingly, HCV seems to evade miR-130a-mediated inhibition by down-regulating its expression, demonstrated in PHHs and Huh7.5.1 cells (Fig. 6d). Significantly lower levels of miR-130a were also observed in liver tissues of CHC patients, compared to healthy controls (Fig. 6e). Of note, hepatic miR-130a expression levels were unrelated to the extent of fibrosis (Supplementary Fig. 11f).

Performing integrative functional, transcriptomics and bioinformatics analyses, as described above for miR-25, we identified six host factors as putative miR-130a targets (Supplementary Fig. 12a, b). These factors are: DDX6, a DEAD box protein expressed in P-bodies and stress granules to mediate translation suppression and RNA degradation; E2F2, a member of the E2F family of transcription factors that are crucial in the cell cycle control; HCCS, a heme-binding, mitochondria-associated protein; INTS6 (DDX26), an RNA helicase involved in transmembrane signaling; LDLR, the low density lipoprotein (LDL) receptor implicated in lipoprotein intake and HCV replication[27, 28]; and NPAT, a histone-specific transcriptional regulator. All six genes contain at least one miR-130 seed match site in their 3′-UTRs (Supplementary Fig. 12c), the activities of which were drastically inhibited by miR-130a transfection (Fig. 6f). The mRNA levels of these genes were also markedly reduced in miR-130a-transfected cells (Fig. 6g). These results confirmed that miR-130a represses the expression of the six host factors. The reliance of HCV

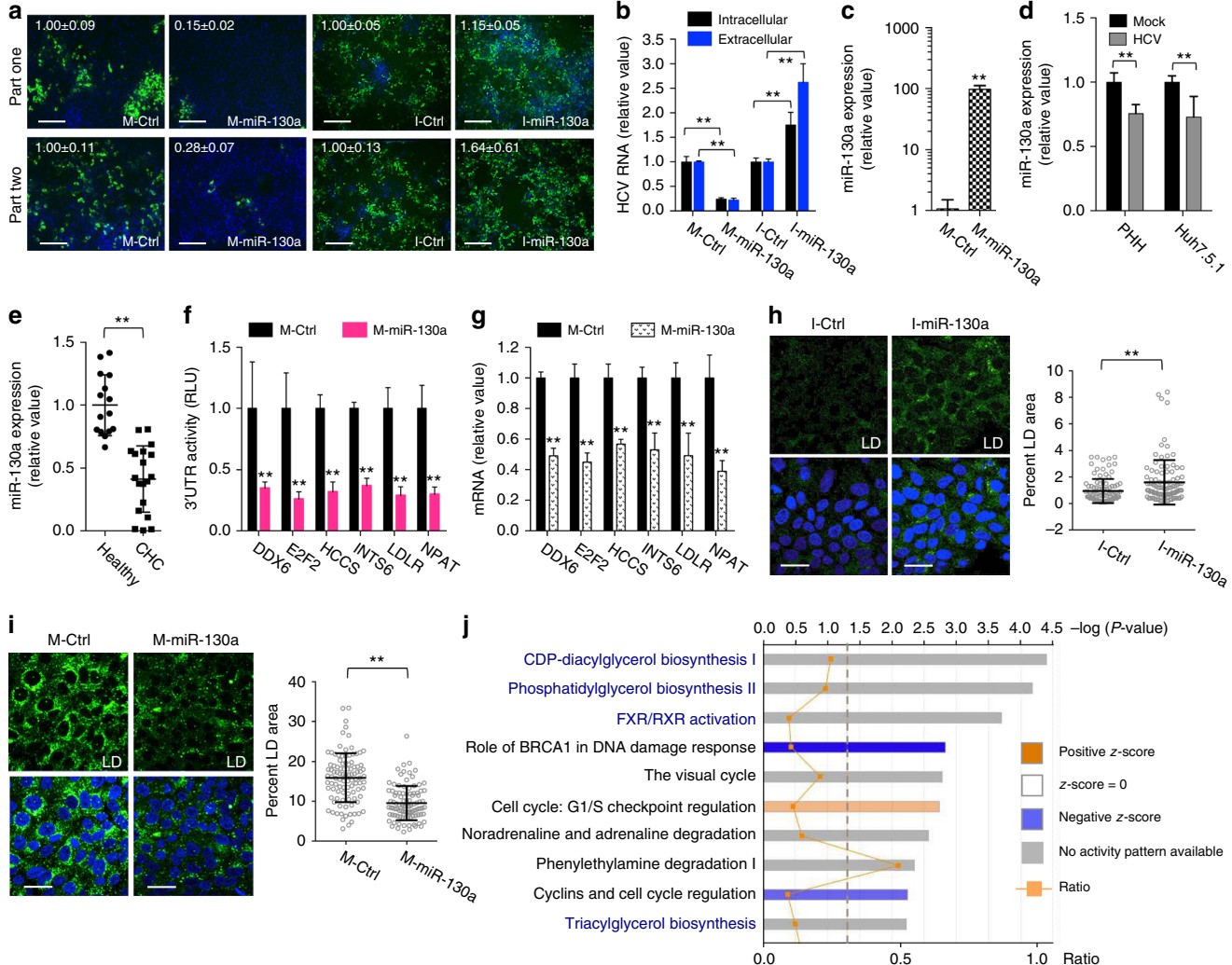

**Fig. 6** Identification of miR-130a that restricts HCV RNA replication and assembly. **a**, **b** miR-130a mimic or inhibitor transfection in Huh7.5.1 cells inhibits or enhances HCV infection, respectively. **a** Part-one and part-two HCV core protein expression were determined by immunostaining. Green, HCV core; blue, nuclei. Scale bars, 100 μm. **b** Levels of intracellular and extracellular HCV RNA were measured by Q-PCR. **c** Transfection efficiency of miR-130a mimic overexpression in Huh7.5.1 cells. **d** HCV infection decreases miR-130a expression in Huh7.5.1 cells and PHH. **e** Comparison of hepatic miR-130a abundance between liver samples of CHC patients and those of healthy controls. **f**, **g** miR-130a mimic transfection inhibits 3'-UTR activities (**f**) and mRNA levels (**g**) of various predicted cellular targets in Huh7.5.1 cells. **h**, **i** Representative images and quantitative analyses of LD contents in Huh7.5.1 cells transfected with control or miR-130a inhibitor (**h**) or mimic (**i**). Green, LDs, stained by BODIPY; blue, nuclei. Scale bars, 100 μm. 100 cells under each indicated condition were counted and percentages of LD area were measured. **i** Cells were treated with oleic acid (40 μM) for 24 h before being stained for LD contents. **j** Bioinformatics-based pathway analysis was conducted for miR-130a downregulated genes (Supplementary Data 7) using IPA. Values are normalized relative to Ctrl, and error bars represent SD of the mean, n = 3 (**a**–**d**, **g**) or 5 (**f**). **P < 0.01 determined by Student's t-test

infection on these miR-130 targets was validated by siRNA-mediated loss-of-function assays. Depletion of each gene in Huh7.5.1 cells specifically restricted HCV RNA replication (Supplementary Fig. 12d, e). Therefore, DDX6, E2F2, HCCS, INTS6, LDLR, and NPAT likely mediate miR-130's inhibitory effect on HCV replication (Supplementary Fig. 12f).

We also investigated the antiviral effect of miR-130b, though it is expressed at a lower level in hepatocytes than miR-130a (Supplementary Fig. 11b). Transfection of miR-130b mimic or hairpin inhibitor significantly decreased or enhanced HCV infection, respectively (Supplementary Fig. 13a). Notably, the inhibitors of less-abundant family members (e.g., miR-130b) exerted less effects than that of the more highly expressed (i.e., miR-130a). This is likely due to the continued function of endogenous miR-130a when the less abundant miR-130b is inhibited. HCV infection also downregulated miR-130b expression in primary hepatocytes and Huh7.5.1 cells (Supplementary

Fig. 13b). Transfecting miR-130b mimic in cells markedly decreased both the 3'-UTR activities and mRNA levels of the six miR-130a-targeted genes, indicating that miR-130b targets the same panel of HCV host dependencies as miR-130a for its antiviral functions (Supplementary Fig. 13c–e).

Noticeably, bioinformatics-based target prediction revealed six other essential HCV host factors—OSBP, PI4KA, RAB10, RABEPK, ROCK2, and VAMP1 that bear miR-130a/b match sites in their 3'-UTRs (Supplementary Fig. 14a). However, these genes were not transcriptionally regulated by miR-130a as shown in the microarray analysis (Supplementary Data 7) and confirmatory qPCR gene expression assay (Supplementary Fig. 14b). We conducted 3'-UTR assay using luciferase reporters and demonstrated that miR-130 mimic had limited effect on their 3'-UTR activities, except ROCK2 showed moderate inhibition (Supplementary Fig. 14c), suggesting that these genes probably are not targets of miR-130.

We noted that HCV assembly/secretion was also affected in miR-130a-sequestered cells. Transfection of miR-130a inhibitor considerably enhanced core protein production in part two of the screen, whereas a modest increase was seen in part one (Fig. 6a). miR-130a inhibition also elevated HCV RNA levels; with a more profound effect exerted on the level of secreted viral RNA (Fig. 6b). We elucidated the mechanism by which miR-130a modulates HCV assembly/secretion. A recent study suggested that miR-130b, an immunometabolism-regulatory miRNA implicated in development of hepatic steatosis, plays a role in lipid metabolism in the liver[29]. Transfection of miR-130b mimic suppressed the expression of various lipogenic genes in hepatocytes, while miR-130b inhibition augmented cellular lipid droplet (LD) contents[30]. Cellular lipogenesis and LD formation are pivotal for HCV assembly[31, 32]. We demonstrated a similar role of miR-130a as transfection of its hairpin inhibitor markedly induced LD contents in Huh7.5.1 cells (Fig. 6h), likely contributing to its role in enhancing HCV assembly. In contrast, treatment of cells with miR-130a mimic drastically reduced oleic acid-triggered LD formation (Fig. 6i). Transcriptome analysis of miR-130a mimic overexpressing cells revealed that multiple cell signaling pathways closely related to LD biogenesis were significantly inhibited (Fig. 6j). The greatest-impaired cellular functions include biosyntheses of phospholipids and triacylglycerol, which constitute LD's phospholipid monolayer and hydrophobic core, respectively[33]. As such, miR-130a alters hepatocellular lipogenic pathways to impede HCV assembly (Supplementary Fig. 12f).

## Discussion

Although appreciable progress has been made in elucidating the regulatory roles of miRNAs in various biological and pathological processes, the interdependencies between pathogens and host miRNAs remain largely unexplored. Previous efforts have suggested that the miRNA machinery mediates variable functions in the infections of many viruses, yet global miRNA-virus interactome analyses are often lacking. The aim of this study is to systematically investigate host miRNome–HCV interactions through integrative functional genomics approaches. We initiated a combined genome-wide miRNA agonist (mimic) and antagonist (hairpin inhibitor) screen and examined the potential impact of each cellular miRNA on HCV propagation. We identified 31 phenotype-specific proviral or antiviral miRNAs, enhancing or restricting HCV infection, respectively. We further dissected the functions and underlying mechanisms of three physiologically relevant miRNA families (miR-25, let-7, and miR-130) in modulating HCV infection. We demonstrated that these miRNAs repress multiple essential cellular co-factors of HCV, thus interfering with various concurrent signal pathways that are pivotal for HCV life cycle. Intriguingly, upon both cell culture-supported HCV infection and chronic infection in hepatitis C patients, the expression of these HCV-restricting miRNAs was significantly downregulated. Our study, by characterizing HCV–miRNA interactions at the whole-genome level, revealed previously unappreciated versatile roles of these RNA molecules in mediating HCV infection and HCV-related pathogenesis.

Cellular miRNAs constitute an important part of HCV host dependencies[5, 34]. miRNAs may modulate HCV infection either by directly targeting viral genome, or indirectly by regulating virus-associated cellular pathways[5]. HCV meanwhile co-opt cellular miRNAs to propagate viral infection and induce continuous liver damage. For example, HCV infection induces the expression of miR-27a/b both in vitro and in vivo[35, 36]. The miR-27 miRNAs target multiple lipid-associated transcription factors to regulate lipid metabolism and LD biogenesis, thus modulating HCV-mediated hepatic steatosis[35, 36]. Altered hepatic lipid metabolism has also been attributed to other miRNAs, including miR-185 and miR-130b, which were shown to be downregulated by HCV. The virus thereby preserves lipid microenvironments that are crucial to the formation of viral replication complex[30]. In addition, two immunoregulatory miRNAs, miR-146a-5p and miR-21-5p, enhance HCV infection and are subject to HCV regulation[37, 38]. miR-146a induction deregulates HCV-associated metabolic pathways, promotes liver inflammation, and accelerates liver disease progression[37]. Aberrant expression of miR-21 triggers HCC growth and metastasis by modulating a PTEN-dependent pathway[39]. These cellular miRNAs represent potential therapeutic targets not only for the treatment of HCV infection but also for the prevention of HCV-mediated liver disease.

It is noteworthy that miR-27a/b, miR-185, miR-130b, and miR-146a-5p mentioned above are all mimic screen hits in our study (Supplementary Data 1). Overexpressing their mimics in hepatocytes considerably altered HCV infection consistent with their previously reported phenotypes[37]. We also showed that miR-21, to a lesser extent, enhances HCV infection (Supplementary Data 1). However, these miRNAs were not identified from the inhibitor screen, likely due to the relatively low hepatic abundance, functional redundancy of other related miRNAs, or the suboptimal design of their hairpin inhibitors that masked the activities in inhibiting the corresponding endogenous miRNAs. As such, these cellular miRNAs were not classified as "validated hits" based on our strict selection criteria but may still be physiologically relevant to HCV infection.

HCV induces proviral miRNAs to evade host immune surveillance and establish persistent infection[37]. For example, HCV upregulates miR-21, which represses MyD88 and IRAK1, two critical components of the TLR pathway, to dampen IFN-α signaling[38]. HCV also enhances hepatic expression of miR-208b and miR-499a-5p, two "myomiRs" encoded in the introns of myosin-encoding genes[40]. These "myomiRs" depress both type I and III IFNs[40, 41]. Nevertheless, the induction of miR-208b or miR-499a-5p by HCV was not observed in our transcriptome analyses. Both miRNAs are barely expressed in either Huh7.5.1 cells or PHHs (Supplementary Data 3 and 4). Whether the miRNA-mediated attenuation of host antiviral immunity is biologically relevant to HCV infection and liver disease pathogenesis remains to be elucidated.

Emerging evidence has suggested that miRNA-mediated gene regulation may play a vital role in the intrinsic antiviral immune response[42–44]. IFNs, the primary defense in mammalian cells, work in concert with cellular miRNAs to curb viral infections[42]. Multiple miRNAs are involved in HCV-associated innate immunity. These include miR-27 and miR-130a/b, which enhance IFN production and the expression of various interferon-stimulated genes (ISGs) in hepatocytes[30, 35, 45]. Several miRNAs, such as let-7b and let-7f, are induced by IFNs[46], and inhibition of their activities will also impair IFN's antiviral effect against HCV[47]. Viruses may hijack antiviral miRNAs to regulate host innate immune responses, affect viral replication in particular cell types and contribute to disease severity[48]. Similarly, HCV might manipulate the IFN-activated miRNAs to restrain replication to a level appropriate for persistent viral infection[34].

miRNA-mediated inhibition can be subverted by viruses during their replication cycle[49–51]. Viruses may interfere with miRNA biogenesis, thus regulating their abundance in cells[34]. Altered expression of cellular miRNAs is regarded as a viral strategy to support its propagation. Recently, a cascade of human miRNAs have been shown to be aberrantly regulated by HCV[14]. Nevertheless, the biological relevance and functional impact of these regulations are not yet defined. Through global miRNA transcriptome analyses, we showed that cellular miRNAs are

broadly downregulated by HCV infection in various hepatocyte-related systems. HCV-mediated downregulation of miRNAs can be attributed to reduced miRNA biogenesis or increased degradation or both, which is worthy of further study. Interestingly, miR-25, miR-130a/b, and let-7a—three most relevant antiviral miRNAs physiologically interacting with HCV—are down-regulated by the virus, demonstrated in both cultured cells and liver tissues of CHC patients. These data suggest that HCV may overcome antiviral miRNA-mediated restriction by attenuating their expression to establish viral persistence.

Diverse biological and pathological processes are subject to miRNA-mediated regulation[4, 22]. Dissecting miRNA targets is critical to uncover their versatile functions and thus unveil human disease mechanisms. Precise prediction of miRNA targets is challenging. Current computational approaches greatly rely on sequence alignment and "seed" matches between miRNAs and their targets[1]. As miRNAs typically decrease mRNA and/or protein levels of their targets, transcriptome and proteomic analyses have been used to obtain more reliable miRNA–target interactions[52]. When miRNAs and targets are simultaneously profiled across different experimental or physiological conditions, they should exhibit inverse expression patterns. Nevertheless, miRNA-mediated inhibition of gene expression is relatively subtle—often causing <50% reduction of mRNA or protein levels. Therefore, interrogating their functions, as conducted in our study, represents a crucial complementary approach to identify physiologically active miRNA targets.

Interestingly, analysis of miRNA–target interactions suggested that HCV-related miRNAs repress the expression of host factors through different mechanisms. While most miRNA targets could be identified by a decrease in mRNA level, there were a few for which translation is suppressed without noticeable mRNA degradation, such as regulation of CLDN1 by let-7a. This divergence in mechanism of target gene inhibition indicates that microarray-based transcriptomics may not be sufficient to uncover the complete set of genes that are regulated by a miRNA. Proteomics that addresses the translational profiling of human genome is necessary to capture all the miRNA-mediated regulatory effects on the expression of target genes.

To further characterize the HCV-related miRNA targets identified from our study, we compared our data with the published miRNA–mRNA interaction database which employed human Argonaute (AGO) crosslinking and immunoprecipitation (AGO-CLIP) and high-throughput sequencing (HiTS) techniques to derive genome-wide miRNA regulatory networks during HCV infection[19]. The majority of the functionally validated targets in our study are confirmed by the HiTS-CLIP, suggesting they are direct targets of their corresponding miRNAs. These include SUV420H1 for miR-25; PPIA, IQCB1, IGF2BP1, and CLDN1 for let-7a; and DDX6, NPAT, LDLR, HCCS, and INTS6 for miR-130a. Nevertheless, the HiTS-CLIP assay, while technically comprehensive in demonstrating direct interaction of miRNA and its target mRNAs, does not capture all the potential miRNA targets. Given the wide variety of factors that influence the expression level of a given gene when a cell is infected, a single-layer analysis of individual mRNAs such as applied by the AGO-CLIP would be inconclusive in evaluating the effects of these miRNAs. Thereby, the identified miRNA–mRNA interactions in our study that are absent in the CLIP database are still likely to be direct miRNA targets based on the various bioinformatics/transcriptomics-derived, and phenotype-based analyses and functional assays performed in the study, although additional studies are necessary to definitively validate this point.

Integrative functional genomics and systems biology approaches have been successfully exploited for global identification of HCV–host interactions[9], and principally can be extended to the elucidation of host miRNome functions and association with HCV infection. Applying these technologies, we uncovered 12 cellular miRNAs as physiologically relevant to HCV infection, and then systematically investigated their effects on HCV life cycle and potential mechanisms. Interrogation of miRNA functions and HCV host dependencies allowed to the elucidation of multiple viral–miRNA–cellular regulatory networks associated with HCV infection. Numerous previously unappreciated direct or indirect connections were identified. This approach revealed the broad impact of cellular miRNAs on the regulation of host responses by HCV. The comprehensive miRNA–mRNA regulatory modules obtained from this study will advance our understanding of HCV–host interactions and disease mechanisms, and potentially yield targets for developing innovative preventive or therapeutic regimens.

## Methods

**Cell line and virus.** The Huh7 derivative cell line Huh7.5.1 (provided by F.V. Chisari of The Scripps Research Institute, La Jolla, CA) were maintained in complete growth medium (DMEM; Corning) containing 10% fetal bovine serum (Corning). The cell line was free from mycoplasma contamination, regularly tested using a Mycoplasma Detection Kit (Life Technologies). HCV JFH-1 strain (provided by T. Wakita of the National Institute of Infectious Diseases, Tokyo, Japan) was propagated and infectivity was titrated as previously described[53–55]. Unless otherwise indicated, HCV infection was conducted at a multiplicity of infection (MOI) of 0.5, and assays were typically performed at 48 h post infection.

**Patients and liver biopsies.** Liver biopsy samples were obtained from patients with CHC (genotype 1b) with or without cirrhosis, who had previously failed a course of peginterferon and ribavirin due to partial virological response (defined as ≥2 log reduction in HCV RNA level compared to baseline by week 12 but with detectable HCV RNA at week 24) or null response (≤2 log reduction in HCV RNA level compared to baseline by week 12) and who were participating in a trial to evaluate asunaprevir 100 mg twice daily and daclatasvir 60 mg once daily for 24 weeks (Bristol-Myers Squibb, Raritan, NJ) (ClinicalTrials.gov: NCT01888900). The initial liver biopsies, obtained within 12 weeks prior to starting therapy, were used in this study. All patients provided written informed consent and the protocol was approved by the Institutional Review Board of the National Institute of Diabetes and Digestive and Kidney Diseases and the National Institute of Arthritis and Musculoskeletal and Skin Diseases.

**miRNA primary screening.** To identify cellular miRNAs that modulate productive HCV infection, we performed a combined, genome-wide miRNA agonist and antagonist screen on arrayed libraries obtained from Dharmacon (miRIDIAN microRNA Mimic and miRIDIAN microRNA Hairpin Inhibitor Libraries) that present ~970 mature miRNAs collected in the miRBase sequence database v.13.0. As in our previous HCVcc siRNA screen[8], we applied a two-part screen that recapitulates both the early and the late stages of HCV life cycle. In "part one", individual miRNA mimics or inhibitors were transfected into Huh7.5.1 cells at a final concentration of 25 nM using a reverse transfection protocol as previously described[8]. Transfection conditions were optimized in a 384-well format (384-well black with clear bottom assay plate, Corning 3712) to obtain the highest level of transfection efficiency without inducing cytotoxicity. Briefly, 0.15 μL (0.5% final concentration) of Oligofectamine (Invitrogen) was diluted in 9.85 μL of Opti-MEM (Gibco) and allowed to incubate at room temperature (RT) for 10 min. This mixture was then distributed to wells (9 μL per well) using a Matrix Wellmate microplate dispenser (Thermo Scientific), and plates were spun down at 250×g for 2 min. Afterwards, the arrayed miRNA mimics or hairpin inhibitors were added manually using multi-channel pipettes (Biohit). For mimics, 1.5 μL were added from a 0.5 μM stock solution and for inhibitors, 1.5 μL were added from a 0.2 μM stock solution. After 20 min, 1000 Huh7.5.1 cells were added robotically into each well in 20 μL of DMEM with 15% FBS. The plates were spun down at 250×g for 5 min, and then placed in a 37 °C humidified incubator with 5% CO₂. The next day, 5 μL of fresh DMEM were added to the outer wells at the plate margins (without miRNA or siRNA transfection) to avoid edge effects. After 72 h of miRNA transfection, the medium was removed with a 24-channel aspirating wand (V&P Scientific), and cells were infected with HCV JFH-1 strain at an MOI of 0.5 in 40 μL of DMEM with 10% FBS. At 48 h post infection, 30 μL of supernatant from each well were transferred to a replica 384-well plate, pre-seeded with 2500 Huh7.5.1 cells per well one day prior, for use in "part two" of the screen. Meanwhile, the "part-one" cells were fixed with 4% paraformaldehyde (Sigma-Aldrich), and then permeabilized in a 0.3% Triton X-100 solution (Sigma-Aldrich, diluted in PBS) containing 3% Albumin Bovine Fraction V (MP Biomedicals) and 10% normal goat serum (Vector Laboratories) for blocking. Cells were subsequently labeled with purified anti-HCV core protein monoclonal antibody (yielded from the anti-core 6G7 hybridoma cells, provided by H. Greenberg and X. He, Stanford

University, Stanford, CA) at a 1:500 dilution in PBS with 1% BSA, and subsequently incubated with Alexa Fluor 488 secondary antibody (Invitrogen) at 1:1000 in PBS with 1% BSA. Cell nuclei were counterstained with Hoechst 33342 (Invitrogen) at 1:5000 in PBS. Each step was followed by three washes with PBS. The cells were then visualized and imaged on an automated scanning microscope (Zeiss) at ×10 magnification, under 488 nm wavelengths to detect Alexa Fluor 488-labeled HCV core and 361 nm wavelengths to detect cellular DNA stained with Hoechst. Images were analyzed using the MetaMorph Microscopy Automation and Image Analysis Software (Molecular Devices) to quantify multiple parameters, including total cell number, and percentage of HCV core-positive cells per well. Two days after viral infection, the "part-two" cells were stained, imaged and quantified under the same procedures as "part-one" cells.

The negative controls, miRIDIAN miRNA mimic negative control #2 (CN-002000-01) or hairpin inhibitor negative control #2 (IN-002005-01), and positive control, hsa-miR-122-5p mimic (C-300591-05) or hairpin inhibitor (IH-300591-06) were included in each plate. ON-TARGETplus APOE siRNA (L-006470-00) was also present in each plate as a positive control for "part-two" HCV infection. In addition, an siRNA pool against Polo like kinase one (siGENOME PLK1 siRNA, M-003290-01) was used in all plates as an indicator of transfection efficiency[8]. All miRNA or siRNA reagents were obtained from Dharmacon. The screening was performed in triplicate, which showed good reproducibility.

Cell viability in each well was also analyzed via imaging of stained cells. Any miRNA that caused substantial cytotoxicity (viable cell number decreased by >50% from the plate mean) was excluded from hit selection and further analysis, based on the criteria established previously[8].

**Rank product statistical analysis and selection of primary screen hits**. For analysis of the primary screen results, $Z$ scores were calculated for miRNA mimic and miRNA hairpin inhibitor, part one and part two data sets independently. Student's $t$-test was used to calculate an overall $P$ value for a given miRNA mimic or inhibitor compared to the entire dataset (e.g., all mimic part one). The $Z$ scores and $P$ values for both mimic and hairpin inhibitor data sets are shown in Supplementary Data 1. For selection of miRNA screen hits, a microRNA must have both a $Z$ score of less than −1 or more than 1 (more than one standard deviation from the mean) and a $P$ value of less than 0.05. Phenotypically confirmed hits were defined as being confirmed hits in both the mimic and inhibitor screens and having opposite phenotypes.

**miRNA secondary screening**. An HCVcc system encoding a Renilla luciferase reporter (HCV P7-Luc, provided by C. Rice of The Rockefeller University, New York, NY) was used for the validation screen. miRNA mimics and hairpin inhibitors cherry-picked from the primary screen were individually transfected into Huh7.5.1 cells grown in 96-well white microplates (Greiner Bio-one). After 72 h, cells were infected with HCV P7-Luc. At 48 h post infection, Renilla luciferase assays (Promega) were performed to determine levels of HCV infection. Secondary screen hits were defined as those miRNAs exhibiting consistent proviral or antiviral phenotypes as observed in the primary screen, and enhancing or inhibiting HCV P7-Luc infection by at least 1.5-fold, with a $P$ value <0.01 (Student's $t$-test).

**miRNA transfection**. miRIDIAN miRNA mimics or hairpin inhibitors (Dharmacon) were transfected into Huh7.5.1 cells at a 25 nM final concentration in 12, 24, or 96-well format. As in the miRNA screening, a reverse transfection protocol using Oligofectamine (Invitrogen) was applied. Typically, at 72 h post-transfection, when miRNA-mediated gene regulation became efficient, cells were either further treated or harvested for various assays.

**shMIMIC lentiviral miRNA infection**. Huh7.5.1 cells were seeded into a 96-well black, clear bottom plate (Corning Incorporated) or 8-well chamber slides (LAB-TEK) at the density of 10,000 or 30,000 cells per well, respectively. After 24 h, the media was replaced with Transduction Medium (DMEM (4.5 g/L glucose, Sodium Pyruvate, 25 mM HEPES, no L-Glut) with 10% FBS, and Polybrene (AmericanBio) at a concentration of 4 µg/mL] combined with either the active control (SMARTvector Non-targeting hCMV-TurboGFP Control), miR-130a (shMIMIC Human Lentiviral microRNA has-miR-130a-3p hCMV-TurboGFP), miR-25 (shMIMIC Human Lentiviral microRNA has-miR-25-3p hCMV-TurboGFP), or let-7a (shMIMIC Human Lentiviral microRNA has-let-7a-5p hCMV-TurboGFP) (GE Dharmacon). The lentiviral particles were added in triplicates at four different MOI: 20, 10, 5, or 2.5 (as described by Dharmacon shMIMIC Lentiviral microRNA protocol). After 48 h, the cells were infected with HCV JFH-1 strain at an MOI of 0.5. Two days after HCV infection, the supernatants from the 96-well plate were transferred to an untreated 96-well plate seeded with Huh7.5.1 cells (25,000 cells per well) for part-two analysis. Immunofluorescent staining using anti-core primary antibody was performed for both part-one and part-two plates, as well as the 8-well chamber slides. For the 96-well plates, images were collected using a high-content scanning microscope, and analyzed using the MetaMorph Microscopy Automation and Image Analysis Software. Multiple parameters, including total cell number, and percentage of HCV core-positive cells per well were quantified. Images from the chamber slides were acquired using a Zeiss LSM 5 Live DuoScan

Confocal Microscope under an oil-immersion 1.4 NA 63× objective lens, and analyzed using the ZEN 2012 software.

**siRNA transfection**. SMARTpool siRNAs (Dharmacon) were transfected into Huh7.5.1 cells at a 50 nM final concentration, using Oligofectamine and a reverse transfection protocol. After 72 h, when silencing efficiency is maximal, further treatments or assays were performed.

**HCV life cycle assays**. HCV life cycle assays were performed to assess the association of each validated miRNA hit with various steps of HCV infection. Specifically, HCVpp (for viral entry), subgenomic replicon (for IRES-mediated translation and RNA replication), HCVsc (for single-cycle infection) and HCVcc (for the entire HCV life cycle) assays were conducted. Detailed protocols for these assays are described below.

**HCV entry assay**. Huh7.5.1 cells were treated with 25 nM of miRNA mimics for three days in 96-well flat bottom white microplates (4500 cells per well, in five replicates), and then infected with HCV pseudoparticles (HCVpp, genotype 1b) or the control pseudovirus VSV-Gpp. HCVpp harboring genotype 1b E1-E2 glycoproteins were derived from pHCV-E1E2.1b3 plasmid (provided by F.L. Cosset of the INSERM U412, Lyon, France), and the pseudovirus was generated as previously described[56, 57]. At 48 h post infection, cells were lysed in 1× Reporter Lysis Buffer (Promega), and firefly luciferase (Promega) activity was subsequently measured using a POLARstar Omega multidetection microplate reader (BMG Labtech). For the assay, miRNA mimics that either increased or decreased pseudoviral entry by more than 50% and $P$ values <0.01 (Student's $t$-test) were considered as proviral or antiviral hits, respectively.

**HCV subgenomic replicon assay**. Huh7.5.1 cells were transfected with various miRNA mimics (at 25 nM) or SMARTpool siRNAs (at 50 nM) in 96-well white plates (in five replicates). After 3 days, cells were further transfected with HCV JFH1-RLuc subgenomic replicon RNA (provided by C. Rice of The Rockefeller University, New York, NY) using DMRIE-C (Invitrogen). Two days later, cell lysates were obtained, and Renilla luciferase (Promega) activity was determined. For the replicon assay, the hit selection criteria are relative Renilla luciferase activity (normalized to mimic or inhibitor control as 1) increased by 1.5-fold (proviral miRNA mimics) or decreased by two-fold (antiviral miRNA mimics), with $P$ values <0.01 (Student's $t$-test).

**HCV IRES-mediated translation assay**. Huh7.5.1 cells were treated with various miRNA mimics (at 25 nM) or SMARTpool siRNAs (at 50 nM) for three days in 96-well white plates (in five replicates), and then transfected with pHCV-CLX-CMV RNA (containing HCV IRES that directs the translation of a firefly luciferase reporter gene, provided by M. Niepmann of Giessen University, Giessen, Germany). At 24 h post transfection, cells were lysed and firefly luciferase activity was subsequently measured. For the IRES assay, the hit selection cutoff points are relative firefly luciferase activity (normalized to mimic or inhibitor control as 1) increased by 1.5-fold (proviral miRNA mimics) or decreased by two-fold (antiviral miRNA mimics), with $P$ values <0.01 (Student's $t$-test).

**Single-cycle HCV infection assay**. The core-defective, assembly deficient single-cycle infectious HCV (HCVsc) was generated from a *trans*-packaging system as previously described[9, 58]. For the assay, Huh7.5.1 cells were transfected with various miRNA mimics at 25 nM for 72 h in 96-well white plates (in five replicates) and then infected with HCVsc. At 48 h post infection, cells were harvested and firefly luciferase activity was measured.

**HCV infectious titer assay**. Huh7.5.1 cells were treated with various miRNA mimics or hairpin inhibitors for 72 h before infection with HCV. At 48 h post infection, viral supernatants were collected and serially diluted by 10-fold in complete growth medium and were subsequently applied to naive Huh7.5.1 cells pre-seeded in 96-well clear bottom black assay plates (Corning) ($1 \times 10^4$ cells per well, in eight replicates). After 48 h, HCV core protein expressed in cells was immunostained. Wells containing at least one core-expressing cell were counted as positive, and the $TCID_{50}$ was calculated based on a previously described method[55].

**NanoString nCounter gene expression assay**. Total RNA samples from Huh7.5.1 cells or PHHs were analyzed according to the manufacturer's instructions for the digital multiplexed NanoString nCounter human v2 miRNA expression assay (NanoString Technologies, Seattle, WA). Briefly, 100 ng of total RNA (under each treatment condition) was prepared by ligating a specific miRNA-tag onto the 3′ end of each mature miRNA, followed by hybridization conducted at 65 °C for 19 h. Samples were subsequently applied to the nCounter™ Preparation Station for automated removal of excess probe and immobilization of probe-transcript complexes on a streptavidin-coated cartridge. Data were collected using the

nCounter™ Digital Analyzer by counting the individual barcodes. NanoString data was normalized using the nSolver Analysis Software.

**miRNA microarray analysis**. Total miRNA from HCV-infected or uninfected Huh7.5.1 cells was extracted using a miRNeasy Mini Kit (Qiagen). The extracted RNA was quantified using a NanoDrop ND-1000 spectrophotometer (Thermo Scientific). The miRNA microarray analysis was then performed by LC Sciences. Briefly, 1 μg of RNA was labeled with Cy3 or Cy5 fluorescent dyes, hybridized, and analyzed by microarray assay on a μParaPlo microfluidics chip capable of detecting miRNA transcripts listed in Sanger miRBase Release 13.0. Multiple control probes were included in each chip. Microarray images were scanned, and numerical intensities were subsequently extracted for control, background, and miRNA probes. For data analysis, background signals were subtracted and data were presented as log-2 scale. Cross-array normalization, t-test, ANOVA, heat map, and SAM analyses were performed.

**Transcriptomics analysis**. Huh7.5.1 cells were transfected with miRNA mimic control or let-7a, miR-130a, or miR-25 mimic at 25 nM for 72 h, in triplicate. Total RNA was then extracted using the RNeasy Kit (Qiagen) according to the manufacturer's instructions. RNA samples were quantified with a NanoDrop spectrophotometer, and the RNA quality was analyzed with an Agilent bioanalyzer (Agilent Technologies). RNA was then amplified with an Enzo kit (Enzo Life Sciences). Amplified complementary RNA (cRNA) was hybridized to an Affymetrix Human 133 Plus 2.0 microarray chip containing 54,675 gene transcripts. Target labeling and hybridization to GeneChips were carried out at the NIDDK Genomics Core Facility. The microarray signals were normalized according to the RMA algorithm. Significantly expressed genes were selected based on ANOVA analysis using Partek Pro software (Partek).

**In vitro transcription of HCV RNA and transfection**. HCV JFH1 and subgenomic replion plasmids were linearized with XbaI (New England Biolabs) and HCV IRES plasmid was linearized with BamHI (New England Biolabs). Linearized DNA was purified by phenol-chloroform-isoamyl alcohol extraction. In vitro transcription was then performed using the MEGAscript T7 kit (Ambion), according to the manufacturer's instructions. The quality and quantity of RNA were evaluated by a NanoDrop ND-1000 spectrophotometer (Thermo Scientific). Aliquots (22 μL, 1 μg/μL) of RNA were stored at −80 °C until further use. RNA transfection was performed in 10 cm dish or 12- or 96-well format using DMRIE-C reagent (Invitrogen) applying a forward transfection protocol.

**Viral RNA isolation and quantification**. Subsequent to various miRNA or siRNA treatments, Huh7.5.1 cells were infected with HCV JFH1 strain for 48 h. Total cellular RNA was then extracted using the RNeasy Mini Kit (Qiagen), and viral RNA from supernatants was isolated using QIAamp Viral RNA Mini Kit (Qiagen). Quantitative RT-PCR (qPCR) was performed to measure intracellular and extracellular HCV RNA copy numbers, using Verso 1-Step RT-qPCR Mix (Life Technologies) on an ABI ViiA 7 Real-Time PCR System (Applied Biosystems). PCR parameters were described previously[8]. HCV primer sequences were: (forward) CGGGAGAGCCATAGTGG, and (reverse) AGTACCACAAGGCCTTT CG (IDT). HCV TaqMan probe used was: 6-FAM CTGCGGAACCGGTGAGTACAC TAMRA (IDT). Human 18S rRNA (Applied Biosystems) was used as the internal control.

**Gene expression assay**. Huh7.5.1 cells were transfected with various miRNA mimics or siRNAs for 72 h and then harvested. Total cellular RNA was prepared using the RNeasy Mini Kit. RNA quality and quantity were assessed on a Nanodrop spectrophotometer. Complementary DNA (cDNA) was then synthesized from total cellular RNA using First-Strand cDNA Synthesis Kit (Roche). The mRNA levels of target genes were subsequently determined by qPCR using gene-specific primers and probes (IDT) and FastStart Universal Probe Master (Roche) on an ABI ViiA 7 Real-Time PCR System. Relative mRNA levels were calculated using the $\Delta\Delta CT$ method, with 18S rRNA or GAPDH RNA (Applied Biosystems) as internal control for normalization.

**Western blotting**. Cells were lysed with RIPA buffer (Sigma) and complete protease inhibitor cocktail (Roche) on ice for 20 min and then spun down at 17,500×g for 20 min. Protein lysates were run on Wes (ProteinSimple) as per manufacturer's instructions. 5 μL of diluted antibody per sample was applied. Multiple antibodies used for western blotting were obtained commercially: Purified mouse anti-human IKK-α (1:40) (Clone B78-1; BD Pharmingen), Claudin-1 (XX7) mouse monoclonal antibody (1:40) (Santa Cruz), E-Cadherin (24E10) rabbit monoclonal antibody (1:40) (Cell Signaling), β-Tubulin monoclonal antibody (1:2000) (TUB 2.1; Sigma). The uncropped images of all western blots in Fig. 5l and Supplementary Fig. 10g were shown in Supplementary Fig. 16.

**Immunofluorescence and confocal microscopy**. shMIMIC lentiviral negative control or lentiviral miRNA TurboGFP-infected (at an MOI of 5) Huh7.5.1 cells grown on Lab-Tek II borosilicate 8-well chamber coverslips (Nunc) were fixed with 4% paraformaldehyde (Sigma-Aldrich) for 15 min, permeabilized in 0.1% Triton X-100 (Sigma-Aldrich), and incubated with blocking solution in PBS containing

3% bovine serum albumin fraction V (BSA, MP Biomedicals) and 10% normal goat serum (Vector Laboratories) for 30 min. Cells were then labeled with anti-core monoclonal antibody diluted (1:500) in PBS with 1% BSA, and subsequently incubated with Alexa Fluor 568 secondary antibody (1:1000) (Life Technologies) in PBS with 1% BSA. Nuclei were counterstained with Hoechst 33342 (Life Technologies) at 1:5000 in PBS. Each step was followed by three washes with PBS. Confocal laser scanning microscopic analysis was performed with an Axio Observer.Z1 microscope equipped with a Zeiss LSM 5 Live DuoScan System under an oil-immersion 1.4 NA ×63 objective lens (Carl Zeiss). Images were acquired using ZEN 2012 software (Carl Zeiss). Dual or triple color images were acquired by consecutive scanning with only one laser line active per scan to avoid cross-excitation.

**Quantitative miRNA real-time PCR assay**. Huh7.5.1 cells and PHH were transfected with various miRNA mimics for 3 days or infected with HCV for indicated amount of time. Cell lysates were vortexed to lyse and biopsies were lysed in TissueLyser LT bead mill (Qiagen). Total miRNA was isolated with a miRNeasy Mini Kit (Qiagen) by manufacturer's instructions. RNA was reverse transcribed using the TaqMan MicroRNA Reverse Transcription Kit (Applied Biosystems) according to the manufacturer's instructions. let-7a, miR-130a, miR-130b, and miR-25 expression levels were determined by qPCR using TaqMan Universal PCR Master Mix (Applied Biosystems) and specific miRNA primers and probes (TaqMan MicroRNA Assays, Applied Biosystems). U6 snRNA was used as an internal control.

**3′-UTR assay**. The LightSwitch 3′-UTR reporter GoClone constructs of various putative miRNA targets were purchased from SwitchGear Genomics (Active Motif). The entire 3′-UTRs were placed downstream of a RenSP luciferase reporter under the control of the cytomegalovirus (CMV) promoter, thus the impact of a miRNA on regulation of transcript stability or translation efficiency could be assessed. For the assay, Huh7.5.1 cells were transfected with miRNA mimic of interest or with mimic control for 24 h in 96-well white plates (in five replicates), and then transfected with 50 ng of GoClone 3′-UTR reporter plasmid using FuGENE 6 transfection reagent (Roche). Two days later, cells were lysed and total luciferase outputs were read using LightSwitch Luciferase Assay Reagent (SwitchGear Genomics).

**ATPlite assay**. Huh7.5.1 cells (10,000 cells per well) were seeded in 96-well white assay plates and then treated with various miRNA mimics or hairpin inhibitors at a final concentration of 25 nM (in five replicates). After 72 h, cells were harvested and lysed with 50 μL of mammalian cell lysis solution (PerkinElmer). After 5 min incubation, 50 μL of ATPlite substrate solution (PerkinElmer) were added, and the luminescence in each well was subsequently measured using a POLARstar Omega multidetection microplate reader. Results were elicited in Supplementary Fig. 15.

**Statistical analysis**. Results are presented as the means ± SD. The two-tailed unpaired Student's t-test was used for statistical analysis. The level of significance is denoted in each figure (*$P < 0.05$, **$P < 0.01$. NS, not significant).

**Data availability**. The data that support the findings of this study are available from the corresponding authors upon reasonable request.

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

## Acknowledgements

We thank L. Hertz, K. Valdez, V. Pène, F. Zhang, and Z. Hu from the Liver Diseases Branch for excellent technical assistance; Y.-Q. Zhang from the National Center for Advancing Translational Sciences (NCATS) for generous help with image acquisition and analysis of part of the screens; NCI NanoString Core Facility and NIDDK Genomics Core for transcriptomics analyses. We also thank Drs. C.M. Rice, F.V. Chisari, T. Wakita, F.-L. Cosset, M. Niepmann, and T. Suzuki for their generosity in providing various reagents. Normal human liver tissues were obtained through the Liver Tissue Cell Distribution System, Minneapolis, Minnesota, which was funded by NIH Contract #HHSN276201200017C. This work was supported by the Intramural Research Program of the National Institute of Diabetes and Digestive and Kidney Diseases, US National Institutes of Health.

## Author contributions

Q.L. and T.J.L. conceived the project and designed the experiments; Q.L., B.L., C.S., S.K., H.A., H.C., S.C., and R.E. performed the experiments; M.G.G. provided human liver biopsy samples. Q.L. and T.J.L. analyzed the data and wrote the manuscript with the input from B.L. and other co-authors.

## Additional information

**Competing interests:** The authors declare no competing financial interests.

