## [Peer Review File · Nature Communications]

Reviewers' comments:

Reviewer #1 (Remarks to the Author):

Using genome-wide miRNA mimetic and inhibitor screens, the authors identified miRNAs that have proviral and antiviral functions in the HCV life cycle. Detailed analyses pointed to roles for identified miRs in the viral life cycle, and several putative host target mRNAs were identified. This is a comprehensive analyses that, unfortunately, suffers from subtle effects of the miRs and the huge number of identified target mRNAs. The only solid effect on HCV entry and viral mRNA translation was described for let7a (Fig. 5). Here effects on viral yield and claudin protein production are impressive. Effects of other miRs are subtle, i.e. two-fold at best (Fig. 3a).

1. While the authors quantitate the amount of miRs in uninfected and infected cells, what is need is the abundance of mimetics per cell. In the absence of this data, it is difficult to judge whether the observations are physiologically relevant.
2. Many target genes were predicted for the identified miRs. Based on bias, a few were tested. While those were regulated by the miRs, effects on HCV were not rigorously tested. For example, did the employed siRNAs cause cell toxicity?
3. Ago-Clip data (Luna et al.) could be used to compare predicted miR-target mRNA interactions in uninfected and infected cells. Maybe this analysis will reveal additional, relevant target mRNAs.
4. The discussion is too long.

Reviewer #2 (Remarks to the Author):

In "Cellular microRNA networks regulate host dependency of hepatitis C virus infection", the authors perform a genome-wide miRNA screen for possible roles in HCV infection. The authors transfected 972 different miRNA mimics, and 970 hairpin inhibitors and affects on HCV viral infection was quantified using HCV core protein expression analysis. As a counter screen supernatants were monitored for late stage viral replication. The authors identified 276 mimics and 153 hairpin inhibitors as 'hits' and of those, 31 miRNAs had opposite phenotypes in both gain and loss of function experiments. Of these, three pro-viral miRNAs and 9 antiviral miRNAs were transcriptionally regulated during infection and their affects on HCV infection confirmed through a HCVcc-Luc infection assay. In particular, the authors identify proviral miR17-5p and antiviral miR-25 as targeting late stage HCV assembly and secretion. In addition, let7a was shown to be associated with host response to HCV infection. Finally, the miR130 family was shown to regulate HCV replication and assembly and the authors identified several possible targets for miR130.

While the manuscript provides an important catalog of miRNAs that can influence HCV

infection, many of the affects reported are rather modest and the biological relevance to clinical disease remains to be validated and explored. More importantly, the text would benefit from judicious editing and careful rewrite for clarity. The authors should address the concerns below prior to publication.

Major concerns:

1. The authors compared the expression of miRs in patients and in healthy individuals in addition to their cell culture work. There is not comment or assessment as to the purity of the samples. i.e. Contamination of other tissues or various cell types could be a concern especially in individuals with clinical disease (where fibrosis and fat content can vary significantly). Normalization to tissue specific house-keeping genes would greatly alleviate such concerns and potential biases in miRNA expression.

Minor concerns:

1. Line 45. The statement that miRNAs plays a role in virus- host interaction requires a reference.
2. Line 384, the inhibitors did not show the opposite effects on virus replication (supplementary results, figure10 a)? Please address it in the text. Transfected mimics for miR130a,130b and 301a led to a decrease in core staining, however their equivalent inhibitors did not increase core staining.
3. The decrease of miR130a was very low in cells (fig. 6d) but a bit bigger in patients (fig. 6e), do the authors have any explanation for this discrepancy?
4. Line 454. The authors should explain their speculation on page 21 that miR130a represses PPAR γ at the post-transcription level to modulate LD synthesis. It is not immediately obvious that their conclusion follows from their observation.
5. Line 84. How many replicates were performed for the screen? The authors should comment about this detail in the text.
6. On lines 162-168, the authors essentially repeat their statements from the previous section.
7. Line 179. Please add reference, after your statement that host microRNAs modulate HCV infection.
8. Line 179. The authors state that "A multitude of host miRNAs that modulate HCV infection in hepatocytes have been uncovered..." Is the statement referring to the data in the manuscript, or previously published data. If the latter, then a reference is required. In addition, the section should be compressed with the previous one for clarity.
9. What do the authors mean in lines 193-194? The authors state that "these results reveal a unique feature of HCV-host interaction through reciprocal regulation...", what do the authors mean by "reciprocal regulation"?
10. Line 198, the authors state that ..."biologically relevant miRNAs", what do the authors mean by "biologically relevant"?
11. Line 204, the authors state that... "productive infection", it is not clear what "productive infection means in the context of an in vitro model.
12. "Pan antiviral"? the authors describe results with 2 viruses (HCV and VSV). The statement of pan antiviral therapy may be a bit of an overreach.
13. Line 223, "...that preferentially target..." should be corrected to read "...as preferentially

targeting...".

14. Line 233, please re- write. A "map" does not "elicit".

15. The text would benefit from the removal of words such as "drastically", "considerably", "barley", "interesting", etc ... (for instance one line 241 and 244, and more).

16. Lines 254-256, there is an association between cellular microRNAs to antiviral effect, however causality has not been demonstrated in the manuscript.

17. Line 268, please change to "Thus, by overlapping the predicted target genes generated from both algorithms, ...".

18. Please specify on line 275, at what time point the cells were transfected with miR-25 mimic? How did the authors set the 1.5x threshold? How many replicates were used in the experiment?

19. Please specify on line 279, by how much miR25 candidate targets were downregulated?

20. Line 289, Not necessarily direct. The observed phenotype could be due to an indirect effect on the target genes.

21. Line 300, are p-values for the enriched pathways FDR corrected? Ingenuity does not necessarily report FDR corrected values.

22. In line 308, "most abundantly" in hepatocytes, relative to other cell types? or relative to other let7 microRNAs?

23. Please explain in line 314, "HCV propagation" what do the authors mean by propagation? Multiple rounds of infection, or something else?

24. Lines 314-316, what is the interpretation of the downregulation of let7 in HCV infected cells (figure 5e,f)? is this mediated by virus, or a host response to infection?

25. Lines 318/340, what is the algorithm? Please specify in the text as it is hard to assess the validity of its application.

26. Line 369, "hairpin inhibitors", please add an "s".

27. Line 371, please substitute "can" to "may".

28. Lines 375-380, please review for clarity.

29. Lines 394-397, please review, the sentence is far too long.

30. Line 411, change "considerably" to "led to"?

31. Lines 421-422, the authors state that "bioinformatics tools revealed..." at what confidence interval? Were other predictions made? How many genes were in the list of predictions? As miRNA's are highly promiscuous, this can be quite relevant.

32. After line 443, please comment on whether cell viability is affected. In fact, throughout the manuscript, there is not mention on the affect of miRNA's tested on cell viability.

Reviewer #3 (Remarks to the Author):

In this manuscript Li et al. describe a comprehensive screen of oligonucleotide miRNA mimic and inhibitors to systematically identify miRNAs that can alter HCV infection. The manuscript describes a lot of work and provides a subset of higher confidence miRNAs that at least indirectly regulate HCV infection as well as partial mechanisms that may account for some of this regulation. Although there is no real complete story regarding virus life cycle or host response to HCV, this large dataset will likely be a useful reference for future studies of HCV and miRNAs.

Critiques:

1. The manuscript is very long and at times the flow of prose is disjointed. Portions of the Discussion are redundant with the Results section. The paragraph in lines 178 – 194 is out-of-place.
2. The studies claiming to identify direct targets of miRNAs lack the important control of mutating the putative docking sites in the 3'UTR targets to show this ablate regulation.
3. The mimic studies are almost certainly performed at super-physiological levels compared to endogenous miRNA expression. No estimate for copy number of the mimics is provided. This concern is muted by the priority given to those miRNAs that score inverse in the mimic/inhibitor screens. Still, the strength of evidence that these miRNAs affect HCV infection (as opposed to off-target effects from synthetic oligonucleotides) would be improved by using DNA vectors-based miRNA expression for a few of the top-studied miRNAs.
4. Although control miR-122 is discussed in depth, consider describing the role of the top miRNAs to emerge from this screen in other viral contexts. For example, the miR-17-93 family has been shown in several previous studies to be pro-viral, perhaps by suppressing the effects of IFN & NFKappa B (PMIDs: 20643939, 24075860, 27512057).

Point-by-point Response

We greatly appreciate the reviewers' insightful and constructive comments regarding our earlier version of the manuscript. We have significantly revised the manuscript per the reviewers' suggestions, and are providing a point-by-point response to the reviewers' comments (in blue) below.

Reviewer #1 (Remarks to the Author):

Using genome-wide miRNA mimetic and inhibitor screens, the authors identified miRNAs that have proviral and antiviral functions in the HCV life cycle. Detailed analyses pointed to roles for identified miRs in the viral life cycle, and several putative host target mRNA were identified. This is a comprehensive analyses that, unfortunately, suffers from subtle effects of the miRs and the huge number of identified target mRNAs. The only solid effect on HCV entry and viral mRNA translation was described for let7a (Fig. 5). Here effects on viral yield and claudin protein production are impressive. Effects of other miRs are subtle, i.e. two-fold at best (Fig. 3a).

Response: We appreciate the reviewer for pointing out an important perspective regarding the biological relevance of the identified HCV-associated cellular miRNAs. In this study, we pursued the truly physiologically relevant miRNAs through multiple strategies. First, we conducted an effective genome-wide miRNA hairpin inhibitor screen (which masks the functions of endogenously expressed miRNAs) and identified a multitude of miRNAs that regulate HCV infection in hepatocytes. The inhibitor screen was then cross-referenced with an unbiased whole-genome miRNA mimic screen, producing 31 miRNAs that exhibited opposite phenotypes in the loss-of-function and gain-of-function assays (see **Fig. 1**). Next, to identify whether these 31 phenotype-specific cellular miRNAs are physiologically relevant to HCV, we elucidated miRNA expression landscapes in Huh7.5.1 cells and primary human hepatocytes (PHHs) by conducting NanoString nCounter and microarray-based transcriptome analyses (see **Suppl. Tabs. 3-5**). Twelve miRNAs that are abundantly expressed in hepatocytes were confirmed to modulate HCV infection in a biologically relevant manner. The study flow is illustrated in **Fig. 2b**. Interestingly, the combined mimic/inhibitor screen and transcriptome analyses also revealed previously unrecognized mechanism of HCV to subvert many of these biologically relevant anti-HCV miRNAs by downregulating their expression in cultured hepatocytes (see **Figs. 2d, e** and **S5**) and livers of chronic hepatitis C patients (see **Figs. 4e, 5f** and **6e**). While some of the effects reported here may not be dramatic (2-fold), all the data are solid and highly significant. The less dramatic effects could also be due to variations in experimental reagents and conditions. Collectively, these findings unveil a unique, intrinsic feature of HCV-host interactions through reciprocal regulation between the virus and associated cellular miRNAs.

1. While the authors quantitate the amount of miRs in uninfected and infected cells,

what is need is the abundance of mimetics per cell. In the absence of this data, it is difficult to judge whether the observations are physiologically relevant.

Response: We thank the reviewer for the constructive advice regarding quantification of the abundance of mimetics per cell. We agree that this measurement is instrumental in defining whether the observed phenotypes of a miRNA are physiologically relevant in hepatocytes. Indeed, the nCounter-based Nanostring analysis is reported in count per cell (see **Tables S3** and **S4**). We apologize for not mentioning it clearly in the previous manuscript. We have stated it explicitly in the revised manuscript.

2. Many target genes were predicted for the identified miRs. Based on bias, a few were tested. While those were regulated by the miRs, effects on HCV were not rigorously tested. For example, did the employed siRNAs cause cell toxicity?

Response: We respectively disagree with the reviewer that the selection of the tested target genes in this study are biased. We applied a combined functional, transcriptomics and bioinformatics-based algorithm that systematically identified a complete list of phenotype-specific targets for each miRNA (miR-25, let-7 and miR-130), which underwent further rigorous validation and analyses. The flow of the algorithm is elicited in **Figs. 4f, S9a** and **S12a** and described in detail in the manuscript (Pages 12~13).

Particularly, these “phenotype-specific miRNA targets” are all previously confirmed HCV host dependencies uncovered from our genome-wide siRNA functional screen (*Li et al., 2009 PNAS*) or other published studies. Their effects on HCV life cycle have been rigorously defined in our recent functional genomics studies (*Li et al., 2014 PLOS Pathogens*). In this study, we conducted multiple HCV life cycle assays and further validated the effects of these miRNA targets on various stages of HCV life cycle (see **Figs. 6d-g, S9d, e, and S12d, e**). The employed siRNAs did not cause appreciable cytotoxicity in these studies as well as in previous functional screens (*Li et al., 2009 PNAS; Li et al., 2014 PLOS Pathogens*).

3. Ago-Clip data (Luna et al.) could be used to compare predicted miR-target mRNA interactions in uninfected and infected cells. Maybe this analysis will reveal additional, relevant target mRNAs.

Response: We appreciate the reviewer for this valuable suggestion. Indeed, we have compared the identified HCV-associated miRNA targets with the published Ago-Clip & HiTS data (*Luna et al., 2015 Cell*), which revealed that the majority of the functionally validated targets in our study are confirmed by the HiTS-CLIP. These include SUV420H1 for miR-25; PPIA, IQCB1, IGF2BP1 and CLDN1 for let-7a; and DDX6, NPAT, LDLR, HCCS and INTS6 for miR-130a. We have mentioned this in the “Discussion” section (see Pages 27~28 of the manuscript).

Nevertheless, individual miR-mRNA interactions cannot be reliably examined applying the Ago-CLIP database, as many other factors may affect the mRNA expression levels of predicted targets during HCV infection. The analysis performed by Luna et al.

examined the predicted targets as sets of genes, and did not confirm individual genes as targets of miR-122. Given the wide variety of factors that influence the expression level of a given gene when a cell is infected, a specific analysis of individual mRNA levels would be inconclusive in evaluating the effects of these miRNAs. As mentioned above, we did use the database qualitatively, to assess whether the miR-mRNA interactions we identified are also found using the Ago-Clip method, and have described the results of this qualitative analysis of the database in the text.

4. The discussion is too long.

Response: We thank the reviewer for pointing out this issue. The discussion section has been significantly condensed and edited for clarity.

Reviewer #2 (Remarks to the Author):

In “Cellular microRNA networks regulate host dependency of hepatitis C virus infection”, the authors perform a genome-wide miRNA screen for possible roles in HCV infection. The authors transfected 972 different miRNA mimics, and 970 hairpin inhibitors and affects on HCV viral infection was quantified using HCV core protein expression analysis. As a counter screen supernatants were monitored for late stage viral replication. The authors identified 276 mimics and 153 hairpin inhibitors as ‘hits’ and of those, 31 miRNAs had opposite phenotypes in both gain and loss of function experiments. Of these, three pro-viral miRNAs and 9 antiviral miRNAs were transcriptionally regulated during infection and their affects on HCV infection confirmed through a HCVcc-Luc infection assay. In particular, the authors identify proviral miR17-5p and antiviral miR-25 as targeting late stage HCV assembly and secretion. In addition, let7a was shown to be associated with host response to HCV infection. Finally, the miR130 family was shown to regulate HCV replication and assembly and the authors identified several possible targets for miR130.

While the manuscript provides an important catalog of miRNAs that can influence HCV infection, many of the affects reported are rather modest and the biological relevance to clinical disease remains to be validated and explored. More importantly, the text would benefit from judicious editing and careful rewrite for clarity. The authors should address the concerns below prior to publication.

Major concerns:

1. The authors compared the expression of miRs in patients and in healthy individuals in addition to their cell culture work. There is not comment or assessment as to the purity of the samples. i.e. Contamination of other tissues or various cell types could be a concern especially in individuals with clinical disease (where fibrosis and fat content can vary significantly). Normalization to tissue specific house-keeping genes would greatly alleviate such concerns and potential biases in miRNA expression.

Response: We thank the reviewer for this insightful suggestion. We have performed two additional analyses to address this concern.

First, we analyzed miRNA expression in patients stratified by Ishak score, an indication of the severity of liver fibrosis. As shown in the new figures of the revised manuscript (see **Figs. S6d, S8f, and S11f**), the presence or extent of liver fibrosis does not affect the expression profiles of miR-130a, let-7a, and miR-25. We hope these data would alleviate the concern that tissue/cell type “impurity” would affect the abundance of hepatic miRNAs.

As suggested by the reviewer, we also examined several hepatocyte-specific markers in the liver tissues of both healthy donors and chronic hepatitis C (CHC) patients, to address the concern that there may be fewer hepatocytes (and more immune cells) in the CHC patient biopsies. We demonstrated that the expression levels of various hepatocyte-specific markers including CYP1A2 and HNF6 are actually slightly higher in the CHC patient biopsies than the healthy donor biopsies (but not significant, see data below). As such, we conclude that the observed HCV-mediated decreased expression of these miRNAs was not due to dilution by other cell types in the liver samples.

Minor concerns:

1. Line 45. The statement that miRNAs plays a role in virus-host interaction requires a reference.

Response: We thank the reviewer for the helpful suggestion. We have added a couple of references that specifically review virus-miRNA interactions.

2. Line 384, the inhibitors did not show the opposite effects on virus replication (supplementary results, figure10 a)? Please address it in the text. Transfected mimics

for miR130a, 130b and 301a led to a decrease in core staining, however their equivalent inhibitors did not increase core staining.

Response: We agree with the reviewer that overexpressing miR-130a or miR-130b hairpin inhibitors did not considerably increase HCV core staining, comparing with the sharp inhibitory effects of their mimic counterparts (**Fig. S11a**). Nevertheless, the core staining data were obtained from the genome-wide screens, which are less quantitative and can generate false-negative results. Hence, we further conducted HCVcc assays by quantifying intracellular and extracellular HCV RNA levels upon miR-130a/b mimic or inhibitor transfection. We found that both miR-130a and miR-130b hairpin inhibitors, when overexpressed in hepatocytes, significantly enhanced HCV infection (see **Figs. 6b** and **S13a**).

Interestingly, inhibition of miR-130a seems to be more efficient than miR-130b inhibition in inducing productive HCV infection. We thus stated in the manuscript that “the effect of inhibitors of less abundant family members (e.g. miR-130b) was less dramatic than that of the more highly expressed family member (i.e. miR-130a). This is likely due to the continued function of the highly expressed miR-130a when the less abundant miR-130b is inhibited.”

In general, transfecting many miRNA hairpin inhibitors in cells exerted less dramatic effects on HCV infection than the effects of their equivalent mimics. We have attributed this difference between miRNA mimics and inhibitors to their different modes of actions, and have addressed this point in various parts of the manuscript (e.g. in the paragraph of Page 7).

3. The decrease of miR130a was very low in cells (fig. 6d) but a bit bigger in patients (fig. 6e), do the authors have any explanation for this discrepancy?

Response: We speculate that there are several factors, such as HCV genotypes, duration of infection, host genetic differences, patient sample variations, etc., that may account for this difference.

4. Line 454. The authors should explain their speculation on page 21 that miR130a represses PPAR γ at the post-transcription level to modulate LD synthesis. It is not immediately obvious that their conclusion follows from their observation.

Response: We thank the reviewer for raising this point. We agree that the speculation that “miR-130a represses PPAR γ expression at the post-transcription level to modulate hepatocellular LD synthesis and thus assembly of HCV” seems arbitrary without further data support. In addition, the PPAR γ part seems redundant and thereby we have removed this part from the revised version of the manuscript.

5. Line 84. How many replicates were performed for the screen? The authors should comment about this detail in the text.

Response: The screen was conducted in triplicate; this was noted in the text.

6. On lines 162-168, the authors essentially repeat their statements from the previous section.

Response: We thank the reviewer for pointing out this issue. We have rewritten the sentences and removed the repeating statements from previous sections.

7. Line 179. Please add reference, after your statement that host microRNAs modulate HCV infection.

Response: As suggested by the reviewer, we have added a reference that excellently summarizes recent advances in addressing HCV-miRNA interdependencies (*Singaravelu et al., 2014 Current Opinion in Virology*).

8. Line 179. The authors state that “A multitude of host miRNAs that modulate HCV infection in hepatocytes have been uncovered...” Is the statement referring to the data in the manuscript, or previously published data. If the latter, then a reference is required. In addition, the section should be compressed with the previous one for clarity.

Response: The statement refers to previously published data. As indicated in the above response, a reference has been added to the revised version of the manuscript.

We appreciate the reviewer for the suggestions to improve the clarity of our manuscript. As advised, the two sections were merged.

9. What do the authors mean in lines 193-194? The authors state that “these results reveal a unique feature of HCV-host interaction through reciprocal regulation...”, what do the authors mean by “reciprocal regulation”?

Response: “Reciprocal regulation” means a mutual regulation between cellular miRNAs and HCV. We have edited the sentence as “These results indicate cellular miRNAs both regulate and are regulated by HCV” for clarity.

10. Line 198, the authors state that ...“biologically relevant miRNAs”, what do the authors mean by “biologically relevant”?

Response: This sentence has been edited to read “12 abundantly expressed miRNA hits identified from the primary screen and transcriptome analyses”.

11. Line 204, the authors state that... “productive infection”, it is not clear what “productive infection means in the context of an in vitro model.

Response: We thank the reviewer for this reminder, and have removed “productive” from the sentence.

12. “Pan antiviral”? the authors describe results with 2 viruses (HCV and VSV). The statement of pan antiviral therapy may be a bit of an overreach.

Response: We appreciate the reviewer for raising this point. The sentence has been edited as: “These miRNAs therefore disrupt HCV entry, and may influence the entry of viruses more broadly” to avoid potential overstatement.

13. Line 223, “...that preferentially target...’ should be corrected to read “...as preferentially targeting...”.

Response: We appreciate the reviewer for the helpful suggestion, and have edited the sentence accordingly.

14. Line 233, please re- write. A “map” does not “elicit”.

Response: We thank the reviewer for pointing out this error, and have changed “eliciting” to “displaying” in the revised version of the manuscript.

15. The text would benefit from the removal of words such as “drastically”, “considerably”, “barley”, “interesting”, etc ... (for instance one line 241 and 244, and more).

Response: The manuscript has been edited based on the constructive suggestion by the reviewer.

16. Lines 254-256, there is an association between cellular microRNAs to antiviral effect, however causality has not been demonstrated in the manuscript.

Response: We have edited the sentence by mentioning the antiviral effects of cellular miRNAs as “a legitimate host restricting strategy” to demonstrate the causality of these miRNA-mediated effects.

17. Line 268, please change to “Thus, by overlapping the predicted target genes generated from both algorithms, ...”.

Response: We appreciate the reviewer for the helpful suggestion. The change has been made in the revised version of the manuscript.

18. Please specify on line 275, at what time point the cells were transfected with miR-25 mimic? How did the authors set the 1.5x threshold? How many replicates were used in the experiment?

Response: We thank the reviewer for asking these questions, which we have answered in the “Online Methods” session under the subtitle of “Transcriptomics analysis”. Basically, cells were transfected with the miRNA mimic for 72 h, before being harvested

for RNA extraction and subjected to microarray analysis. The experiments were performed in triplicate. A 1.5-fold of change plus *P* value less than 0.05 was considered significant, according to the protocol previously established (*Feld et al., 2007 Hepatology*).

19. Please specify on line 279, by how much miR25 candidate targets were downregulated?

Response: As suggested by the reviewer, we have specified in the text that “two out of twenty-seven” putative miR-25 targets were downregulated.

20. Line 289, Not necessarily direct. The observed phenotype could be due to an indirect effect on the target genes.

Response: We agree with the reviewer that the observed effects on target 3'UTRs might due to an “indirect” effect or other forms of regulation. Therefore, we have removed the term “direct” from the sentence.

Nevertheless, we believe that the binding of miR-25 seed region to various 3'UTRs of its putative targets most likely exerts a direct regulatory effect, as the isolation of the 3'UTR from the promoter and coding region dramatically reduces the likelihood that the regulation we are seeing is due to some other mechanisms (e.g. changes in transcription).

21. Line 300, are p-values for the enriched pathways FDR corrected? Ingenuity does not necessarily report FDR corrected values.

Response: The *P* values for the enriched pathways shown in **Figs. 4i** and **6j** all represent FDR corrected values analyzed by Ingenuity.

22. In line 308, “most abundantly” in hepatocytes, relative to other cell types? or relative to other let7 microRNAs?

Response: This sentence has been reworded to make clear that let-7a is the most abundantly expressed let-7 family member.

23. Please explain in line 314, “HCV propagation” what do the authors mean by propagation? Multiple rounds of infection, or something else?

Response: HCV propagation means viral replication. We have edited this sentence to read “HCV infection” to avoid confusion.

24. Lines 314-316, what is the interpretation of the downregulation of let7 in HCV infected cells (figure 5e,f)? is this mediated by virus, or a host response to infection?

Response: The downregulation of let-7 expression in HCV-infected cells may result from reduced miRNA biogenesis or increased degradation or both. It is either a virus-mediated effect to attenuate the antiviral activity of let-7, or a host stress response to viral infection. While the question is of great interest, we believe it is beyond the scope of this paper which is already data-dense and complex.

25. Lines 318/340, what is the algorithm? Please specify in the text as it is hard to assess the validity of its application.

Response: We used the same algorithm to identify let-7a targets as that was used to access miR-25 targets. The flow of this combined bioinformatics, phenotype and transcriptomics-based algorithm is explained in detail in pages 12~13. To avoid redundancy, we have mentioned that “we applied the same algorithm that was employed for the identification of miR-25 targets” in the revised manuscript. For let-7a, a summary of the algorithm used, including the number of genes identified at each step, is shown in **Fig. S9a**.

26. Line 369, “hairpin inhibitors”, please add an “s”.

Response: The change has been made in the revised manuscript.

27. Line 371, please substitute “can” to “may”.

Response: We thank the reviewer for the helpful suggestion, and have edited this sentence accordingly.

28. Lines 375-380, please review for clarity.

Response: We have rewritten the sentences in this section to enhance clarity.

29. Lines 394-397, please review, the sentence is far too long.

Response: We thank the reviewer for this reminder. We have revised the long sentence and divided it to several short sentences to increase readability.

30. Line 411, change “considerably” to “led to”?

Response: The change has been made, according to the reviewer’s suggestion.

31. Lines 421-422, the authors state that “bioinformatics tools revealed...” at what confidence interval? Were other predictions made? How many genes were in the list of predictions? As miRNA’s are highly promiscuous, this can be quite relevant.

Response: The methods used to identify these putative targets are described in detail above in the manuscript (see miR-25 target prediction and verification); for brevity, they

are not described again in this section. An overview of this process for miR-130a, along with the number of genes identified, can be found in **Fig. S12a**.

32. After line 443, please comment on whether cell viability is affected. In fact, throughout the manuscript, there is not mention on the affect of miRNA's tested on cell viability.

Response: We thank the reviewer for this constructive comment. We have described the tested miRNA's effects on cell viability (see **Fig. S15**) at multiple places of the revised manuscript.

Reviewer #3 (Remarks to the Author):

In this manuscript Li et al. describe a comprehensive screen of oligonucleotide miRNA mimic and inhibitors to systematically identify miRNAs that can alter HCV infection. The manuscript describes a lot of work and provides a subset of higher confidence miRNAs that at least indirectly regulate HCV infection as well as partial mechanisms that may account for some of this regulation. Although there is no real complete story regarding virus life cycle or host response to HCV, this large dataset will likely be a useful reference for future studies of HCV and miRNAs.

Response: We greatly appreciate the reviewer's positive comments about the comprehensiveness, quality and reliability of our data and manuscript.

Critiques:

1. The manuscript is very long and at times the flow of prose is disjointed. Portions of the Discussion are redundant with the Results section. The paragraph in lines 178 – 194 is out-of-place.

Response: We agree with the reviewer that the earlier version of the manuscript was too complex and data-dense, and the flow of prose is sometimes hard to follow. In the revised manuscript, we have substantially edited and streamlined various text parts, particularly the "Discussion" section, for a more concise, clear, coherent and logic body of work.

2. The studies claiming to identify direct targets of miRNAs lack the important control of mutating the putative docking sites in the 3'UTR targets to show this ablate regulation.

Response: We thank the reviewer for raising this point regarding validation of miRNA targets through mutagenesis studies. Indeed, in our manuscript, the targets of three miRNAs investigated in depth (miR-25, let-7, and miR-130) were systematically identified and verified through extensive bioinformatics/transcriptomics-derived, and

phenotype-based analyses. Furthermore, by performing various functional assays, including 3'UTR activity assays, qRT-PCR and Western blotting for target gene expression, and proviral/antiviral effects on HCV infection, we have validated these bona fide targets in a more convincing manner. These assays are typically used in the literature to validate the direct targets of miRNAs. While mutagenesis analyses on the 3'UTRs would be helpful, they would require a large amount of additional work, which seems to be excessive, and may increase the complexity of the manuscript, which is already data-dense.

3. The mimic studies are almost certainly performed at super-physiological levels compared to endogenous miRNA expression. No estimate for copy number of the mimics is provided. This concern is muted by the priority given to those miRNAs that score inverse in the mimic/inhibitor screens. Still, the strength of evidence that these miRNAs affect HCV infection (as opposed to off-target effects from synthetic oligonucleotides) would be improved by using DNA vectors-based miRNA expression for a few of the top-studied miRNAs.

Response: We highly appreciate the reviewer for the constructive suggestion. We agree that using DNA vector-based miRNA overexpression systems would further confirm the intimate roles of the identified cellular miRNAs that are physiologically relevant to HCV infection. As such, we have conducted additional experiments by infecting Huh7.5.1 cells with shMIMIC lentiviral miRNAs (GE Dharmacon) including miR-25, let-7a, or miR-130a. The miRNA expressing cells enabled more precise and sensitive evaluation of miRNA-induced gene target modulation and phenotypic effects. We showed that the lentiviral vector-based miR-25, let-7a or miR-130a expression in Huh7.5.1 cells, as expected, significantly inhibited HCV infection, in a lentiviral MOI-dependent manner (see Supplementary **Figs. 6b, c, 8c, d, and 11d, e** of the revised version of the manuscript).

4. Although control miR-122 is discussed in depth, consider describing the role of the top miRNAs to emerge from this screen in other viral contexts. For example, the miR-17-93 family has been shown in several previous studies to be pro-viral, perhaps by suppressing the effects of IFN & NFKappa B (PMIDs: 20643939, 24075860, 27512057).

Response: We thank the review for this insightful perspective regarding the potential roles of HCV-associated miRNAs in other viral contexts. In light of the versatile functions that the cellular miRNA machinery exerts in viral infections and virus-host interactions, it is fairly confident to speculate that some of the identified HCV-modulating miRNAs may also impact the infection of other viruses, such as the effects of miR-17-93 family through inhibition of innate immunity as the reviewer mentioned. Indeed, several other miRNA hits identified from our screen have also been shown to regulate other viral infections by previously published studies. Nevertheless, due to space limit, we will not be able to address these points in the current manuscript, which is already complex and data-dense.

Reviewers' comments:

Reviewer #1 (Remarks to the Author):

The authors have carefully addressed previously raised concerns. The is clearly an important data set that should be published. Some known microRNA-mRNA were verified and novel interactions were identified.

Reviewer #2 (Remarks to the Author):

While the manuscript could still benefit from judicious editing, as it seems quite long, we thank the authors for the changes that they have made to the manuscript. They have greatly improved the readability as well as scientific impact of the work.

Reviewer #3 (Remarks to the Author):

The authors were partially responsive to my original critiques but my major critique still applies.

Major critique:

For those target transcripts based on luciferase 3' UTR reporter assays and lacking CLIP verification, the authors cannot rule out that the effects they observe are indirect. That is, what rules out that these miRNAs targets an unknown factor that participates in 3'UTR gene regulation as opposed to directly regulating these transcripts? The standard for the field is to use point mutant 3' UTR reporters showing ablation of regulation. Without verification, those mRNA targets of miRNA deemed "direct" throughout this manuscript are overstated and possibly wrong. Either the appropriate controls need to be done or all statements of "direct" need to be tempered.

Minor critique:

The Discussion section is still rather long. The following text seems to be an overstatement: "we systematically investigated the roles of all HCV-related cellular miRNAs in 588 the entire viral life cycle and explored their potential mechanisms."

Minor problem:

Unless this is some sort of control that I am not understanding and I missed in the text, I believe the Figures in the Supplemental section using lentiviral constructs to express miRNAs are mislabeled "miR-25" in the top left panel of Fig. S6,8,11. Rather, these should be the appropriate miRNA being exogenously expressed.

Point-by-point Response

We highly appreciate the favorable comments of the reviewers regarding our revised manuscript. We are providing a point-by-point response to Reviewer #3's specific comments (in blue) below.

Reviewer #1 (Remarks to the Author):

The authors have carefully addressed previously raised concerns. There is clearly an important data set that should be published. Some known microRNA-mRNA were verified and novel interactions were identified.

Reviewer #2 (Remarks to the Author):

While the manuscript could still benefit from judicious editing, as it seems quite long, we thank the authors for the changes that they have made to the manuscript. They have greatly improved the readability as well as scientific impact of the work.

Reviewer #3 (Remarks to the Author):

The authors were partially responsive to my original critiques but my major critique still applies.

Major critique:

For those target transcripts based on luciferase 3' UTR reporter assays and lacking CLIP verification, the authors cannot rule out that the effects they observe are indirect. That is, what rules out that these miRNAs target an unknown factor that participates in 3'UTR gene regulation as opposed to directly regulating these transcripts? The standard for the field is to use point mutant 3' UTR reporters showing ablation of regulation. Without verification, those mRNA targets of miRNA deemed "direct" throughout this manuscript are overstated and possibly wrong. Either the appropriate controls need to be done or all statements of "direct" need to be tempered.

Response: We thank the reviewer for raising this point regarding validation of miRNA targets through data mining the published AGO-CLIP database or performing 3'UTR mutagenesis studies. While we agree that the point mutant 3'UTR reporter data would confirm some of the miRNA-mRNA interactions identified in our study as being the "direct" targets, we believe these assays are unlikely to yield a meaningful outcome to our study, which is already overly data-laden. The main findings of our study are the systematic and genome-wide identification of miRNAs involved in the regulation of HCV infection and the elucidation of their mechanisms of action by functionally and bioinformatically linking to host mRNAs and their encoded proteins that are known to be host dependency factors for HCV. We would like to mention that many of the mRNAs identified to be the targets of the miRNAs are verified by the published AGO-CLIP database. Nevertheless, due to an inevitable caveat of the HiTS-CLIP assay – false negatives, it does not capture all the potential miRNA targets. In addition, individual

miRNA–mRNA interactions cannot be comprehensively and reliably examined applying the AGO-CLIP database only, as many other factors may affect the mRNA expression levels of predicted targets during HCV infection. Given the wide variety of factors that influence the expression level of a given gene when a cell is infected, a single-layer analysis of individual mRNA levels such as applied by AGO-CLIP would be inconclusive in evaluating the effects of these miRNAs. Thereby, the identified miRNA–mRNA interactions in our study that are not present in the CLIP database are still likely to be direct miRNA targets based on the various bioinformatics/transcriptomics-derived, and phenotype-based analyses and functional assays performed in this study.

The reviewer pointed out that it is possible that the miRNA may be directly regulating an unknown host factor that targets the 3'UTR of the mRNA for regulation in an “indirect” manner. While this may be true, it does not mitigate the fact that the miRNA still functionally regulates the mRNA of question, whether directly or indirectly. We appreciate the reviewer for not insisting that mutating the 3'UTRs is necessary for this already voluminous study to be accepted, and suggesting that instead we can temper the statement of “direct” regulation in the manuscript. We have thus modified the manuscript to tone down the conclusion regarding “direct” regulation.

Minor critique:

The Discussion section is still rather long. The following text seems to be an overstatement:

“we systematically investigated the roles of all HCV-related cellular miRNAs in 588 the entire viral life cycle and explored their potential mechanisms.”.

Response: We thank the reviewer for pointing out this error, and have rephrased the sentence as “we systematically investigated the roles of all 12 physiologically relevant HCV-associated cellular miRNAs ...” in the Discussion section.

Minor problem:

Unless this is some sort of control that I am not understanding and I missed in the text, I believe the Figures in the Supplemental section using lentiviral constructs to express miRNAs are mislabeled “miR-25” in the top left panel of Fig. S6,8,11. Rather, these should be the appropriate miRNA being exogenously expressed.

Response: We apologize for this typo, and have corrected the labeling of let-7a and miR-130a in Fig. S8 and Fig. S11, respectively.